# Improving the Transferability of Adversarial Attacks through Experienced Precise Nesterov Momentum

## Abstract

Deep Neural Networks are vulnerable to adversarial attacks, which makes adversarial attacks serve as a method to evaluate the robustness of DNNs. However, adversarial attacks have high white-box attack success rates but poor transferability, making black-box attacks impracticable in the real world. Momentum-based attacks were proposed to accelerate optimization to improve transferability. Nevertheless, conventional momentum-based attacks accelerate optimization inefficiently during early iterations since the initial value of momentum is zero, which leads to unsatisfactory transferability. Therefore, we propose Experienced Momentum (EM), which is the pre-trained momentum. Initializing the momentum to EM can help accelerate optimization during the early iterations. Moreover, the pre-update of conventional Nesterov momentum based attacks is rough, prompting us to propose Precise Nesterov momentum (PN). PN refines the pre-update by considering the gradient of the current data point. Finally, we integrate EM with PN as Experienced Precise Nesterov momentum (EPN) to further improve transferability. Extensive experiments against normally trained and defense models demonstrate that our EPN is more effective than conventional momentum in the improvement of transferability. Specifically, the attack success rates of our EPN-based attacks are $\sim$11.9% and $\sim$13.1% higher than conventional momentum-based attacks on average against normally trained and defense models, respectively.

## 1 Introduction

Deep neural networks (DNNs) (Krizhevsky et al., 2012; Szegedy et al., 2015; He et al., 2016; Ioffe & Szegedy, 2015) have been widely applied in computer vision, e.g., autonomous driving (Franchi et al., 2022; Hao et al., 2019; Cococcioni et al., 2018), facial recognition (Chrysos et al., 2020; Ghenescu et al., 2018), and medical image analysis (Akselrod-Ballin et al., 2016; Ding et al., 2017; Liu et al., 2019). However, Szegedy et al. (2013) found that applying certain imperceptible perturbations to images can make DNNs misclassify, and they refer to such perturbed images as *adversarial examples (AEs)*. Adversarial examples pose a huge threat to the security of DNNs, which attaches extensive attention from researchers.

Adversarial attacks can be categorized into white-box attacks and black-box attacks. Typically, iterative gradient-based (Kurakin et al., 2016; Madry et al., 2017) and optimization-based attacks (Carlini & Wagner, 2017) have high white-box but low black-box attack success rates, which means that such two attacks are impracticable in the real world. Transferability, which means adversarial examples crafted on the source model remain effective on other models, makes black-box attacks feasible. Furthermore, iterative gradient-based attacks have the advantages of low computational cost and fast generation speed, thus improving the transferability of iterative gradient-based attacks has become a hotspot in the field of adversarial attacks.

Many methods have been proposed to improve the transferability of iterative gradient-based attacks. These methods can be classified into three branches: improving optimization algorithms, input transformations, and disrupting feature space. For example, MI-FGSM (Dong et al., 2018), NI-FGSM (Lin et al., 2019), and VM(N)I-FGSM (Wang & He, 2021) improve gradient ascent (or

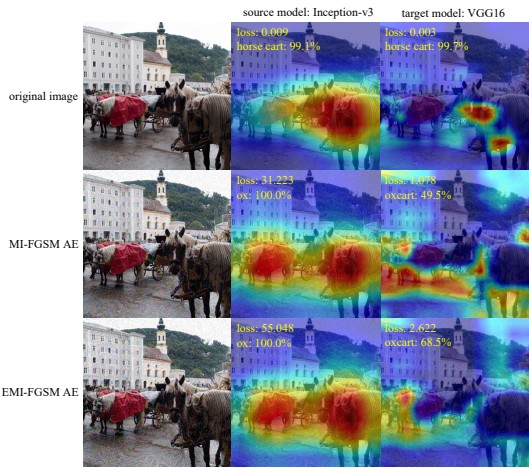

Figure 1: Comparison of conventional Polyak momentum and experienced Polyak momentum. Adversarial examples are crafted on the source model (Inception-v3) and used to attack the target model (VGG16). Our Experienced MI-FGSM (EMI-FGSM), which integrates EM into MI-FGSM, causes misclassification with higher loss and confidence than MI-FGSM, thus EMI-FGSM can mislead the attention of the target model better than MI-FGSM.

descent) algorithm to escape from saddle points and poor local extrema to improve transferability; DIM (Xie et al., 2019), TIM (Dong et al., 2019), and SIM (Lin et al., 2019) craft adversarial examples on a set of models derived by input transformations to prevent overfitting and improve transferability; NRDM (Naseer et al., 2018), FDA (Ganeshan et al., 2019), and FIA (Wang et al., 2021) disrupt deep features of DNNs to craft highly transferable adversarial examples.

Those mentioned above adversarial attacks mostly adopt the momentum (Polyak, 1964; Nesterov, 1983) to accelerate optimization. However, such momentum-based adversarial attacks (e.g., M(N)I-FGSM, VM(N)I-FGSM, and FIA) have the problem of initializing the momentum to zero, resulting in inefficient acceleration due to momentum accumulating few gradients during the first few iterations. Therefore, we propose Experienced Momentum (EM), which is the pre-trained momentum. Before the iterations, the momentum is initialized to EM instead of zero, leading to better acceleration in the first few iterations. The comparison of conventional Polyak momentum (Polyak, 1964) and experienced Polyak momentum is shown in Fig. 1. To prevent overfitting on the source model, we train EM on a set of models derived by Random Channels Swapping (RCS). EM and RCS are detailed in Sec. 3.1.

Furthermore, adversarial attacks (e.g., NI-FGSM and VNI-FGSM) based on Nesterov momentum (i.e., Nesterov Accelerated Gradient, NAG (Nesterov, 1983)) have the disadvantage that the pre-update is rough. Specifically, during each iteration, the parameters are first pre-updated along the momentum to obtain the pre-update point, which is an estimation of the next position. Then the pre-update is modified by the gradient of the pre-update point. Such looking-ahead property of Nesterov momentum makes parameters escape from saddle points and poor local extrema easier and faster, resulting in improving transferability. However, pre-updating only along the momentum is rough, and the estimation of the next position of the parameters is imprecise. Therefore, we propose Precise Nesterov momentum (PN), which not only retains the looking-ahead property but also refines the pre-update by adopting the gradient of the current data point. To improve transferability further, we integrate EM with PN as Experienced Precise Nesterov momentum (EPN). PN and EPN are detailed in Sec. 3.2.

Overall, we make the following contributions:

- We propose Experienced Momentum (EM), which is trained on a set of models derived by Random Channels Swapping (RCS). Initializing the momentum to EM can accelerate optimization effectively during the early iterations to improve transferability.

- We propose Precise Nesterov momentum (PN), which adopts the gradient of the current data point to refine the pre-update to escape from saddle points and poor local extrema easier and faster. We also integrate EM with PN as Experienced Precise Nesterov momentum (EPN) to improve transferability further.
- Extensive experiments on normally trained and defense models demonstrate that our EPN is more effective than conventional momentum for improving transferability.

## 2  RELATED WORK

### 2.1  TRANSFERABLE ADVERSARIAL ATTACKS

Since adversarial examples were discovered by Szegedy et al. (2013), many methods (Goodfellow et al., 2014; Kurakin et al., 2016; Carlini & Wagner, 2017) have been proposed to craft adversarial examples to demonstrate the vulnerability of DNNs. We focus on the transferability of iterative gradient-based attacks and review related works from three branches: improving optimization algorithms, input transformations, and disrupting feature space.

**Improving optimization algorithms.** Dong et al. (2018) integrated Polyak momentum (Polyak, 1964) into I-FGSM (Kurakin et al., 2016) to accelerate gradient ascent (or descent) to improve transferability. Inspired by the fact that Nesterov momentum (Nesterov, 1983) is superior to Polyak momentum, Lin et al. (2019) integrated Nesterov momentum into I-FGSM to improve transferability further. Wang & He (2021) used the gradient variance of the previous iteration to tune the current gradient to stabilize the update direction and escape from saddle points and poor local extrema.

**Input transformations.** The nature of input transformations is crafting adversarial examples on a set of derived models to prevent overfitting. Xie et al. (2019) performed random resizing and padding with probability $p$ to derive models. Dong et al. (2019) convolved the gradient to approximate translating input. Lin et al. (2019) scaled the input with the scale factor $1/2^i$ to derive a set of models.

**Disrupting feature space.** Naseer et al. (2018) created maximum distortions in the feature space to craft adversarial examples, based on the intuition that features of DNNs are highly generalizable. Ganeshan et al. (2019) highly corrupted deep features by disrupting features at each layer of DNNs to improve transferability. Wang et al. (2021) described feature importance with the aggregate gradient and disrupted important object-aware features to achieve stronger transferability.

### 2.2  ADVERSARIAL TRAINING

Adversarial training as a common defense measure can validate transferability further. Adversarial training increases robustness by adding adversarial examples to the training data. Goodfellow et al. (2014) showed that adversarially trained models are more robust. However, Kurakin et al. (2016) pointed out that adversarial training is not robust to iterative attacks. Moreover, Tramèr et al. (2017) showed that adversarially trained models are still vulnerable to simple white-box and black-box attacks. Therefore, they proposed *ensemble adversarial training* adding adversarial examples crafted from other models to the training data.

## 3  METHODOLOGY

Given a target model $f'(\boldsymbol{x}; \boldsymbol{\theta}')$, where $\boldsymbol{x}$ is an input, and $\boldsymbol{\theta}'$ is the parameters of $f'$. Let $J(\cdot, y)$ be a loss function, where $y$ is the ground-truth label of the input $\boldsymbol{x}$. A non-targeted adversarial example $\boldsymbol{x}^{adv}$ satisfies $f'(\boldsymbol{x}; \boldsymbol{\theta}') \neq f'(\boldsymbol{x}^{adv}; \boldsymbol{\theta}')$ under the constraint of $||\boldsymbol{x}^{adv} - \boldsymbol{x}||_p \leq \epsilon$, where $|| \cdot ||_p$ denotes the $L^p$ norm, and $p$ is generally $0, 1, 2, \infty$. In this paper, we focus on $p = \infty$. Note that our methods can be generalized to $p = 0, 1, 2$ easily. Crafting non-targeted adversarial examples can be described as solving the following optimization problem:

$$\arg\max_{\boldsymbol{x}^{adv}} J(f'(\boldsymbol{x}^{adv}; \boldsymbol{\theta}'), y), \quad \text{s.t. } ||\boldsymbol{x}^{adv} - \boldsymbol{x}||_p \leq \epsilon. \tag{1}$$

In this paper, we focus on non-targeted attacks. Our proposed methods can be easily transformed into targeted attacks by replacing the above objective function with $-J(f'(\boldsymbol{x}^{adv}; \boldsymbol{\theta}'), y^*)$, where $y^*$ denotes the target label.

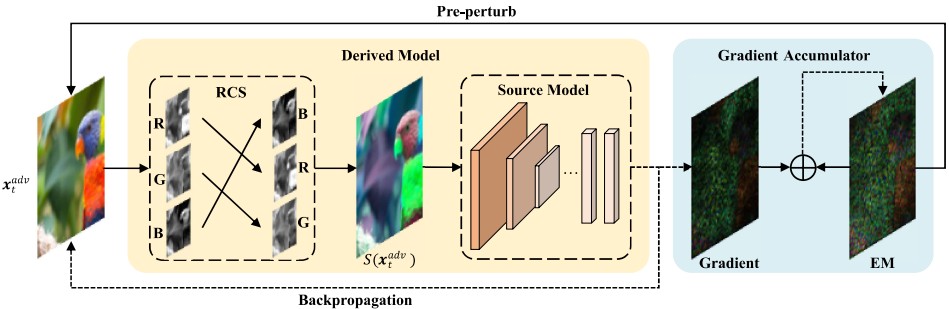

Figure 2: Illustration of training EM during each iteration.

Many gradient-based methods have been proposed to solve Eq. 1, e.g., FGSM (Goodfellow et al., 2014), I-FGSM (Kurakin et al., 2016), and PGD (Madry et al., 2017). However, the parameters $\boldsymbol{\theta}'$ of the target model $f'$ is inaccessible for black-box attacks, resulting in the inability to solve Eq. 1 directly. Therefore, the target model $f'$ is usually replaced with a model $f$ (i.e., the source model) with accessible parameters $\boldsymbol{\theta}$, and then adversarial examples are crafted on the source model $f$ to attack the target model $f'$. To achieve effective black-box attacks, adversarial examples crafted on the source model $f$ are required to have high transferability. Therefore, we propose Experienced Momentum (EM, detailed in Sec. 3.1) and Precise Nesterov momentum (PN, detailed in Sec. 3.2) to improve transferability. EM and PN can be naturally combined as Experienced Precise Nesterov momentum (EPN, detailed in Sec. 3.2) to further improve transferability.

## 3.1 EXPERIENCED MOMENTUM

Momentum-based attacks initialize momentum to zero, resulting in inefficient acceleration during the first few iterations. Therefore, we propose Experienced Momentum (EM), which is the pre-trained momentum. Setting the initial momentum to EM can accelerate the optimization during the early iterations. To prevent overfitting of EM and improve transferability further, we train EM on a set of models derived by Random Channels Swapping (RCS). RCS derives models by randomly swapping the channels of the input image, which is equivalent to randomly swapping the "block" dimensions of the original model, leading to various decision boundaries of derived models. Therefore, training EM on derived models can prevent overfitting. The specific procedure for training EM is as follows.

First of all, we perform RCS on the input image $\boldsymbol{x}$. Specifically, we denote the input image $\boldsymbol{x}$ as an RGB triplet $(R, G, B)$, and then the input image $\boldsymbol{x}$ through RCS can be denoted as $S(\boldsymbol{x})$, where $S(\boldsymbol{x}) \in \{(R, G, B), (R, B, G), (G, R, B), (G, B, R), (B, R, G), (B, G, R)\}$, $S(\cdot)$ denotes RCS. Secondly, $S(\boldsymbol{x})$ is fed into the source model $f$ to derive $f(S(\cdot), y)$. Thirdly, we pre-perturb the input image $\boldsymbol{x}$ on the derived model $f(S(\cdot), y)$ by iterative gradient-based attacks to prevent overfitting. As shown in Fig. 2, we accumulate gradients to training EM during each iteration. Finally, we follow the above procedure repeatedly to make the EM more generalizable. After training EM, we set the initial value of momentum to EM to accelerate the early iterations.

## 3.2 PRECISE NESTEROV MOMENTUM

Nesterov momentum based Attack (e.g., NI-FGSM (Lin et al., 2019) and VNI-FGSM (Wang & He, 2021)) only pre-update along the momentum roughly, resulting in the imprecision of the pre-update point that is the estimate of the next iterative position. Against this disadvantage, we propose Precise Nesterov momentum (PN), which considers the gradient of the current data point in the pre-update to make the pre-update precise. Specifically, during each iteration, the pre-update is performed along the gradient of the current data point and momentum successively to obtain the pre-update point, and then we use the gradient of the pre-update point to modify the pre-update. We integrate PN into I-FGSM as PNI-FGSM. The $t$-th iteration of PNI-FGSM can be formalized as follows:

$$\widetilde{\boldsymbol{x}}_t^{adv} = \boldsymbol{x}_t^{adv} + \alpha \cdot \left[ \frac{\nabla_{\boldsymbol{x}_t^{adv}} J(f(\boldsymbol{x}_t^{adv}; \boldsymbol{\theta}), y)}{||\nabla_{\boldsymbol{x}_t^{adv}} J(f(\boldsymbol{x}_t^{adv}; \boldsymbol{\theta}), y)||_1} + \mu \cdot \boldsymbol{g}_{t-1} \right], \tag{2}$$

---

**Algorithm 1:** Experienced Precise Nesterov momentum I-FGSM (EPNI-FGSM)

---

**Input** : A source model $f$ with parameters $\boldsymbol{\theta}$ and a loss function $J$. An original image $\boldsymbol{x}$ with ground-truth label $y$.
**Input** : The maximum perturbation $\epsilon$, the number of iterations $T$, and the decay factor $\mu$.
**Input** : The epochs of pretraining $epochs$.
**Output:** An adversarial example $\boldsymbol{x}^{adv}$.

1   $\alpha \leftarrow \epsilon/T; \boldsymbol{g}^{exp} \leftarrow \boldsymbol{0}$;
2   **for** $n \leftarrow 1$ **to** $epochs$ **do**
3      $\hat{\boldsymbol{x}}_1^{adv} \leftarrow \boldsymbol{x}$ ;
4      **for** $t \leftarrow 1$ **to** $T$ **do**
5          $\widetilde{\boldsymbol{x}}_t^{adv} \leftarrow \hat{\boldsymbol{x}}_t^{adv} + \alpha \cdot \left[ \dfrac{\nabla_{\hat{\boldsymbol{x}}_t^{adv}} J(f(S(\hat{\boldsymbol{x}}_t^{adv});\boldsymbol{\theta}),y)}{||\nabla_{\hat{\boldsymbol{x}}_t^{adv}} J(f(S(\hat{\boldsymbol{x}}_t^{adv});\boldsymbol{\theta}),y)||_1} + \mu \cdot \boldsymbol{g}^{exp} \right]$ ;
6          $\boldsymbol{g}^{exp} \leftarrow \dfrac{\nabla_{\hat{\boldsymbol{x}}_t^{adv}} J(f(S(\hat{\boldsymbol{x}}_t^{adv});\boldsymbol{\theta}),y)}{||\nabla_{\hat{\boldsymbol{x}}_t^{adv}} J(f(S(\hat{\boldsymbol{x}}_t^{adv});\boldsymbol{\theta}),y)||_1} + \mu \cdot \boldsymbol{g}^{exp} + \dfrac{\nabla_{\widetilde{\boldsymbol{x}}_t^{adv}} J(f(\widetilde{\boldsymbol{x}}_t^{adv};\boldsymbol{\theta}),y)}{||\nabla_{\widetilde{\boldsymbol{x}}_t^{adv}} J(f(\widetilde{\boldsymbol{x}}_t^{adv};\boldsymbol{\theta}),y)||_1}$ ;
7          $\hat{\boldsymbol{x}}_{t+1}^{adv} \leftarrow \text{Clip}_{(\boldsymbol{x},\epsilon)} \left\{ \hat{\boldsymbol{x}}_t^{adv} + \alpha \cdot \text{sign}(\boldsymbol{g}^{exp}) \right\}$ ;
8      **end**
9   **end**
10   $\boldsymbol{x}_1^{adv} \leftarrow \boldsymbol{x}; \boldsymbol{g}_0 \leftarrow \boldsymbol{g}^{exp}$ ;
11   **for** $t \leftarrow 1$ **to** $T$ **do**
12      Update $\boldsymbol{g}_t$ and $\boldsymbol{x}_{t+1}^{adv}$ by Eq. 2, 3, 4;
13   **end**
14   **return** $\boldsymbol{x}^{adv} \leftarrow \boldsymbol{x}_{T+1}^{adv}$.

---

Table 1: The abbreviations used in the paper.

| Abbreviation | Explanation |
|---|---|
| D(T)I-MI-FGSM | the combination of D(T)IM and MI-FGSM |
| SI-NI-FGSM | the combination of SIM and NI-FGSM |
| D(T,S)I-EPNI-FGSM | the combination of D(T,S)IM and EPNI-FGSM |
| VT-M(N)I-FGSM | i.e., VM(N)I-FGSM |
| VT-EPNI-FGSM | the combination of Variance Tuning (VT) (Wang & He, 2021) and EPNI-FGSM |
| FI-MI-FGSM | i.e., FIA |
| FI-EPNI-FGSM | the combination of Feature Importance-aware (FI) (Wang et al., 2021) and EPNI-FGSM |

$$\boldsymbol{g}_t = \frac{\nabla_{\boldsymbol{x}_t^{adv}} J(f(\boldsymbol{x}_t^{adv};\boldsymbol{\theta}),y)}{||\nabla_{\boldsymbol{x}_t^{adv}} J(f(\boldsymbol{x}_t^{adv};\boldsymbol{\theta}),y)||_1} + \mu \cdot \boldsymbol{g}_{t-1} + \frac{\nabla_{\widetilde{\boldsymbol{x}}_t^{adv}} J(f(\widetilde{\boldsymbol{x}}_t^{adv};\boldsymbol{\theta}),y)}{||\nabla_{\widetilde{\boldsymbol{x}}_t^{adv}} J(f(\widetilde{\boldsymbol{x}}_t^{adv};\boldsymbol{\theta}),y)||_1}, \tag{3}$$

$$\boldsymbol{x}_{t+1}^{adv} = \text{Clip}_{(\boldsymbol{x},\epsilon)} \left\{ \boldsymbol{x}_t^{adv} + \alpha \cdot \text{sign}(\boldsymbol{g}_t) \right\}, \tag{4}$$

where $\boldsymbol{g}_t$ denotes the momentum, $\boldsymbol{g}_0 = \boldsymbol{0}$, and $\mu$ denotes the decay factor.

We combine EM and PN as Experienced Precise Nesterov momentum (EPN) to further improve transferability. The algorithm of EPNI-FGSM, which integrates EPN into I-FGSM, is summarized in Algorithm 1. Particularly, if $\frac{\nabla_{\widetilde{\boldsymbol{x}}_t^{adv}} J(f(\widetilde{\boldsymbol{x}}_t^{adv};\boldsymbol{\theta}),y)}{||\nabla_{\widetilde{\boldsymbol{x}}_t^{adv}} J(f(\widetilde{\boldsymbol{x}}_t^{adv};\boldsymbol{\theta}),y)||_1} = \boldsymbol{0}$, EPNI-FGSM degrades to Experienced MI-FGSM (EMI-FGSM). If $\frac{\nabla_{\hat{\boldsymbol{x}}_t^{adv}} J(f(S(\hat{\boldsymbol{x}}_t^{adv});\boldsymbol{\theta}),y)}{||\nabla_{\hat{\boldsymbol{x}}_t^{adv}} J(f(S(\hat{\boldsymbol{x}}_t^{adv});\boldsymbol{\theta}),y)||_1} = \boldsymbol{0}$, EPNI-FGSM degrades to Experienced NI-FGSM (ENI-FGSM). If $epochs = 0$, EPNI-FGSM degrades to PNI-FGSM.

## 4 EXPERIMENTS

We conduct extensive experiments on normally trained and defense models to validate that our EPN is more efficient than conventional momentum. We first present the experimental settings in Sec. 4.1. Then, we report the results for attacking normally trained and defense models in Sec. 4.2 and

Table 2: The attack success rates (%) of adversarial examples crafted on source models against normally trained target models. "*" indicates the model being white-box attacked. "Avg" means the average attack success rate.

| Models | Attacks | Iv1 | Iv3 | Iv4 | IRv2 | R18 | R34 | R50 | R101 | R152 | V11 | V13 | V16 | V19 | D121 | D169 | D201 | D161 | Avg |
|---|---|---|---|---|---|---|---|---|---|---|---|---|---|---|---|---|---|---|---|
| | MI-FGSM | 53.8 | 100.0* | 49.6 | 42.5 | 44.0 | 43.4 | 46.4 | 39.5 | 39.9 | 52.1 | 54.4 | 53.9 | 52.1 | 45.2 | 43.3 | 39.5 | 38.5 | 49.3 |
| | NI-FGSM | 64.3 | 100.0* | 59.5 | 53.0 | 53.0 | 49.7 | 52.7 | 46.4 | 45.8 | 60.8 | 62.6 | 63.6 | 62.9 | 51.9 | 49.8 | 45.4 | 48.4 | 57.0 |
| | EPNI-FGSM (Ours) | 79.7 | 100.0* | 74.9 | 68.7 | 63.9 | 65.5 | 67.6 | 59.9 | 60.4 | 70.8 | 73.0 | 74.3 | 72.2 | 68.3 | 65.7 | 63.6 | 64.5 | 70.2 |
| | DI-MI-FGSM | 70.8 | 99.9* | 66.2 | 60.9 | 61.3 | 62.0 | 64.2 | 58.0 | 56.7 | 65.1 | 69.2 | 67.9 | 68.2 | 66.2 | 63.0 | 61.4 | 61.2 | 66.0 |
| | TI-MI-FGSM | 41.8 | 99.7* | 37.9 | 31.5 | 46.4 | 47.6 | 44.9 | 41.9 | 38.6 | 54.9 | 56.7 | 56.2 | 58.3 | 47.9 | 46.5 | 43.5 | 42.8 | 49.2 |
| Iv3 | SI-NI-FGSM | 77.4 | 100.0* | 72.7 | 69.8 | 67.6 | 67.3 | 68.0 | 61.5 | 62.1 | 70.0 | 72.8 | 73.8 | 71.9 | 70.7 | 68.4 | 64.8 | 65.9 | 70.9 |
| | DI-EPNI-FGSM (Ours) | 88.7 | 100.0* | 87.3 | 86.0 | 78.5 | 81.3 | 83.4 | 80.2 | 80.5 | 80.4 | 83.8 | 83.8 | 84.2 | 86.1 | 84.9 | 82.9 | 83.4 | 84.4 |
| | TI-EPNI-FGSM (Ours) | 70.6 | 100.0* | 68.8 | 59.6 | 69.3 | 69.8 | 68.9 | 65.5 | 63.3 | 73.5 | 76.5 | 78.5 | 77.9 | 74.7 | 72.9 | 70.8 | 71.5 | 72.5 |
| | SI-EPNI-FGSM (Ours) | 90.2 | 100.0* | 87.1 | 85.1 | 80.1 | 81.2 | 81.7 | 76.4 | 78.2 | 80.6 | 83.1 | 84.4 | 82.6 | 86.7 | 83.8 | 80.9 | 81.7 | 83.8 |
| | VT-MI-FGSM | 70.5 | 100.0* | 69.7 | 64.9 | 61.7 | 63.6 | 65.7 | 60.0 | 58.6 | 65.0 | 69.2 | 68.7 | 66.7 | 65.8 | 63.8 | 65.0 | 62.6 | 67.1 |
| | VT-NI-FGSM | 76.2 | 100.0* | 76.2 | 70.5 | 65.6 | 68.4 | 71.2 | 65.2 | 64.1 | 70.5 | 73.4 | 73.4 | 73.2 | 71.2 | 71.3 | 69.2 | 67.2 | 72.2 |
| | VT-EPNI-FGSM (Ours) | 84.8 | 100.0* | 83.3 | 78.1 | 77.0 | 78.8 | 79.4 | 75.8 | 73.3 | 80.2 | 83.3 | 82.0 | 81.5 | 79.7 | 80.1 | 78.3 | 78.5 | 80.8 |
| | FI-MI-FGSM | 85.8 | 97.1* | 85.4 | 81.8 | 79.4 | 80.1 | 79.5 | 76.3 | 74.5 | 81.0 | 82.7 | 82.7 | 83.2 | 81.3 | 78.8 | 77.2 | 77.8 | 81.4 |
| | FI-EPNI-FGSM (Ours) | 90.0 | 97.4* | 88.0 | 84.8 | 81.0 | 82.4 | 84.3 | 79.6 | 78.5 | 85.5 | 87.0 | 85.9 | 86.3 | 84.0 | 83.5 | 80.5 | 81.7 | 84.7 |
| | MI-FGSM | 46.4 | 44.0 | 45.4 | 99.6* | 40.0 | 39.2 | 41.3 | 33.9 | 32.7 | 49.9 | 53.2 | 51.4 | 50.6 | 38.6 | 35.3 | 35.0 | 32.7 | 45.2 |
| | NI-FGSM | 49.0 | 45.6 | 48.5 | 100.0* | 40.6 | 38.7 | 42.1 | 34.9 | 32.9 | 53.5 | 54.3 | 54.6 | 54.3 | 37.5 | 37.5 | 33.8 | 33.2 | 46.5 |
| | EPNI-FGSM (Ours) | 68.1 | 65.3 | 68.4 | 100.0* | 55.2 | 53.8 | 55.5 | 48.4 | 47.9 | 63.8 | 67.8 | 66.6 | 66.6 | 54.7 | 50.7 | 49.0 | 48.1 | 60.5 |
| | DI-MI-FGSM | 63.2 | 62.3 | 65.4 | 98.4* | 54.2 | 54.2 | 55.7 | 48.6 | 51.0 | 59.3 | 64.4 | 62.8 | 64.7 | 57.0 | 52.8 | 51.8 | 52.6 | 59.9 |
| | TI-MI-FGSM | 41.5 | 44.5 | 42.0 | 97.9* | 47.1 | 47.6 | 47.3 | 42.9 | 41.8 | 55.9 | 56.7 | 55.5 | 53.9 | 49.2 | 45.3 | 46.9 | 41.9 | 50.5 |
| IRv2 | SI-NI-FGSM | 74.7 | 81.7 | 75.1 | 99.3* | 65.4 | 65.8 | 69.7 | 65.4 | 64.7 | 68.9 | 71.6 | 71.8 | 73.0 | 72.1 | 68.7 | 67.3 | 68.0 | 72.0 |
| | DI-EPNI-FGSM (Ours) | 82.8 | 83.2 | 84.1 | 100.0* | 70.7 | 72.3 | 74.4 | 67.5 | 71.1 | 74.6 | 79.5 | 78.2 | 79.6 | 75.0 | 73.0 | 69.6 | 70.9 | 76.9 |
| | TI-EPNI-FGSM (Ours) | 65.1 | 68.2 | 67.6 | 99.9* | 66.6 | 69.5 | 68.9 | 65.2 | 63.2 | 73.8 | 75.0 | 73.6 | 74.2 | 72.9 | 68.4 | 69.1 | 65.4 | 71.0 |
| | SI-EPNI-FGSM (Ours) | 90.9 | 93.1 | 88.6 | 100.0* | 79.1 | 81.3 | 85.5 | 79.6 | 82.9 | 83.2 | 85.5 | 86.0 | 86.1 | 86.3 | 83.6 | 81.5 | 84.6 | 85.8 |
| | VT-MI-FGSM | 63.2 | 66.6 | 68.8 | 99.6* | 57.8 | 57.8 | 60.7 | 53.6 | 54.9 | 61.7 | 63.5 | 64.4 | 64.0 | 59.8 | 56.4 | 54.2 | 56.1 | 62.5 |
| | VT-NI-FGSM | 66.3 | 71.0 | 73.1 | 99.8* | 59.3 | 59.3 | 62.3 | 56.9 | 56.9 | 66.8 | 68.2 | 68.2 | 68.1 | 60.9 | 58.0 | 57.2 | 58.5 | 65.3 |
| | VT-EPNI-FGSM (Ours) | 81.0 | 82.1 | 83.1 | 100.0* | 71.7 | 72.4 | 75.4 | 70.0 | 70.6 | 77.3 | 80.4 | 79.9 | 78.7 | 77.3 | 73.4 | 72.0 | 71.9 | 77.5 |
| | FI-MI-FGSM | 76.3 | 76.0 | 76.3 | 89.7* | 67.5 | 68.0 | 69.7 | 66.8 | 65.5 | 72.5 | 73.7 | 73.4 | 73.5 | 70.7 | 68.3 | 66.8 | 67.0 | 71.9 |
| | FI-EPNI-FGSM (Ours) | 78.0 | 77.3 | 78.0 | 91.0* | 69.8 | 69.9 | 72.2 | 68.9 | 67.4 | 73.5 | 74.0 | 74.5 | 73.6 | 71.7 | 69.0 | 66.9 | 67.2 | 73.1 |
| | MI-FGSM | 68.9 | 59.4 | 53.8 | 49.4 | 81.0 | 83.3 | 92.1 | 94.5 | 100.0* | 72.7 | 74.1 | 72.5 | 72.5 | 86.5 | 83.0 | 82.5 | 84.0 | 77.1 |
| | NI-FGSM | 75.7 | 64.1 | 58.0 | 51.7 | 86.0 | 87.7 | 94.7 | 96.9 | 100.0* | 76.1 | 77.4 | 77.0 | 76.2 | 87.4 | 85.1 | 85.5 | 86.7 | 80.4 |
| | EPNI-FGSM (Ours) | 91.5 | 85.8 | 79.2 | 74.2 | 94.8 | 96.2 | 98.7 | 99.4 | 100.0* | 88.7 | 89.8 | 89.5 | 89.6 | 96.5 | 96.5 | 97.6 | 96.7 | 92.0 |
| | DI-MI-FGSM | 85.6 | 82.9 | 75.5 | 72.1 | 91.8 | 93.8 | 96.0 | 96.8 | 100.0* | 84.0 | 84.4 | 85.5 | 84.6 | 93.7 | 94.0 | 93.7 | 94.4 | 88.8 |
| | TI-MI-FGSM | 57.1 | 51.3 | 50.1 | 41.4 | 70.9 | 75.0 | 80.5 | 84.5 | 100.0* | 65.4 | 65.0 | 64.1 | 64.0 | 75.1 | 71.5 | 70.7 | 68.1 | 67.9 |
| R152 | SI-NI-FGSM | 85.1 | 77.2 | 71.1 | 66.5 | 89.6 | 92.0 | 95.3 | 97.8 | 100.0* | 82.1 | 83.0 | 81.9 | 81.9 | 93.6 | 91.8 | 92.4 | 91.8 | 86.7 |
| | DI-EPNI-FGSM (Ours) | 97.6 | 96.9 | 94.2 | 92.1 | 98.0 | 98.9 | 99.4 | 99.6 | 100.0* | 94.3 | 94.6 | 94.7 | 95.2 | 99.4 | 99.3 | 99.6 | 99.1 | 97.2 |
| | TI-EPNI-FGSM (Ours) | 80.3 | 78.2 | 75.7 | 66.8 | 89.3 | 91.4 | 94.0 | 95.5 | 100.0* | 81.7 | 81.2 | 80.8 | 81.5 | 91.8 | 89.2 | 91.0 | 89.7 | 85.8 |
| | SI-EPNI-FGSM (Ours) | 95.3 | 91.4 | 88.2 | 84.8 | 96.7 | 97.6 | 98.6 | 99.0 | 100.0* | 90.3 | 91.2 | 92.2 | 90.5 | 99.0 | 98.2 | 98.1 | 98.2 | 94.7 |
| | VT-MI-FGSM | 83.8 | 79.6 | 74.1 | 70.9 | 92.2 | 93.7 | 96.4 | 97.5 | 100.0* | 83.6 | 82.8 | 84.4 | 83.8 | 94.2 | 92.4 | 93.1 | 93.6 | 88.0 |
| | VT-NI-FGSM | 87.7 | 81.9 | 78.9 | 74.4 | 93.5 | 94.9 | 98.2 | 98.7 | 100.0* | 85.3 | 87.2 | 88.2 | 86.9 | 96.1 | 94.9 | 95.1 | 95.7 | 90.4 |
| | VT-EPNI-FGSM (Ours) | 95.4 | 92.1 | 89.7 | 86.9 | 97.9 | 98.8 | 99.5 | 99.8 | 100.0* | 93.6 | 93.5 | 93.7 | 94.4 | 99.4 | 98.7 | 99.4 | 98.8 | 96.0 |
| | FI-MI-FGSM | 93.3 | 88.6 | 88.7 | 85.1 | 95.5 | 96.8 | 97.5 | 98.9 | 99.9* | 92.5 | 92.0 | 93.0 | 93.8 | 97.0 | 95.7 | 96.6 | 96.9 | 94.2 |
| | FI-EPNI-FGSM (Ours) | 95.4 | 93.2 | 93.1 | 90.1 | 96.8 | 97.9 | 98.3 | 98.9 | 100.0* | 94.1 | 94.8 | 95.0 | 95.1 | 97.6 | 97.4 | 97.3 | 97.6 | 96.0 |
| | MI-FGSM | 78.0 | 59.2 | 63.8 | 43.8 | 80.0 | 73.3 | 75.0 | 63.7 | 58.7 | 94.7 | 98.3 | 99.8* | 99.2 | 77.0 | 69.5 | 65.3 | 64.9 | 74.4 |
| | NI-FGSM | 78.8 | 61.6 | 68.8 | 47.3 | 82.3 | 76.1 | 78.0 | 67.0 | 60.4 | 96.8 | 99.2 | 99.9* | 99.1 | 79.1 | 72.1 | 66.4 | 66.0 | 76.4 |
| | EPNI-FGSM (Ours) | 93.5 | 82.3 | 87.0 | 68.1 | 91.5 | 90.0 | 90.0 | 82.9 | 77.6 | 99.0 | 100.0 | 100.0* | 99.9 | 91.2 | 87.3 | 85.8 | 87.0 | 89.0 |
| | DI-MI-FGSM | 88.4 | 75.6 | 77.5 | 59.5 | 87.8 | 83.9 | 86.2 | 73.8 | 69.7 | 98.3 | 98.8 | 100.0* | 99.4 | 81.5 | 78.2 | 77.5 | | 83.8 |
| | TI-MI-FGSM | 60.6 | 51.1 | 48.5 | 33.4 | 71.2 | 65.4 | 60.3 | 53.6 | 48.4 | 88.1 | 92.9 | 99.8* | 94.4 | 64.9 | 58.0 | 54.8 | 52.4 | 64.6 |
| V16 | SI-NI-FGSM | 89.8 | 77.5 | 80.7 | 62.5 | 89.1 | 84.8 | 85.9 | 77.1 | 73.0 | 98.5 | 99.6 | 100.0* | 100.0 | 87.8 | 81.9 | 78.9 | 80.2 | 85.1 |
| | DI-EPNI-FGSM (Ours) | 96.2 | 90.6 | 92.6 | 81.9 | 95.9 | 94.1 | 93.5 | 88.6 | 85.5 | 99.8 | 99.9 | 100.0* | 100.0 | 95.6 | 93.1 | 93.0 | 91.4 | 93.6 |
| | TI-EPNI-FGSM (Ours) | 82.5 | 74.6 | 77.4 | 58.1 | 88.4 | 84.2 | 82.7 | 75.9 | 70.4 | 96.8 | 98.2 | 100.0* | 98.1 | 86.1 | 80.7 | 78.8 | 77.4 | 83.0 |
| | SI-EPNI-FGSM (Ours) | 96.1 | 89.1 | 91.8 | 80.0 | 93.6 | 91.2 | 92.5 | 86.4 | 84.8 | 99.4 | 100.0 | 100.0* | 100.0 | 95.6 | 92.6 | 91.0 | 92.2 | 92.7 |
| | VT-MI-FGSM | 87.5 | 75.2 | 77.9 | 61.9 | 89.9 | 86.1 | 86.8 | 78.3 | 74.1 | 97.7 | 98.9 | 99.9* | 99.4 | 87.8 | 82.2 | 81.3 | 80.6 | 85.0 |
| | VT-NI-FGSM | 89.8 | 76.9 | 80.7 | 65.6 | 92.1 | 87.5 | 89.3 | 81.1 | 76.1 | 98.7 | 99.5 | 99.9* | 99.4 | 88.7 | 85.4 | 82.5 | 82.9 | 86.8 |
| | VT-EPNI-FGSM (Ours) | 95.6 | 88.4 | 92.4 | 81.1 | 96.1 | 94.4 | 94.0 | 90.4 | 88.8 | 99.4 | 99.9 | 99.9* | 99.9 | 95.6 | 93.7 | 93.2 | 92.2 | 93.8 |
| | FI-MI-FGSM | 95.9 | 89.1 | 93.1 | 79.7 | 95.6 | 94.9 | 93.4 | 90.4 | 87.1 | 99.6 | 99.8 | 100.0* | 99.8 | 94.3 | 91.5 | 90.2 | 88.6 | 93.1 |
| | FI-EPNI-FGSM (Ours) | 96.7 | 90.4 | 94.1 | 84.3 | 96.7 | 95.9 | 96.0 | 92.6 | 90.1 | 99.8 | 99.8 | 100.0* | 99.9 | 96.4 | 94.3 | 93.3 | 93.0 | 94.9 |

Sec. 4.3, respectively. Finally, we provide ablation studies in Sec. 4.4. Table 1 introduces the abbreviations used in the paper.

## 4.1 EXPERIMENTAL SETTINGS

**Dataset.** We follow the previous works (Dong et al., 2019; Wang et al., 2021) to use the DEV dataset from the NIPS17 Adversarial Attacks and Defenses Competition. This dataset contains 1000 images with size $299 \times 299$.

**Target Models.** Seventeen normally trained models, i.e., GoogLeNet (Iv1) (Szegedy et al., 2015), Inception-v3 (Iv3) (Szegedy et al., 2016), Inception-v4 (Iv4), Inception-ResNet-v2 (IRv2) (Szegedy et al., 2017), ResNet-18 (R18), ResNet-34 (R34), ResNet-50 (R50), ResNet-101 (R101), ResNet-152 (R152) (He et al., 2016), VGG11 (V11), VGG13 (V13), VGG16 (V16), VGG19 (V19) (Simonyan & Zisserman, 2014), DenseNet-121 (D121), DenseNet-169 (D169), DenseNet-201 (D201), and DenseNet-161 (D161) (Huang et al., 2017). Ten defense models (i.e., adversarially trained models), i.e., Adv-Inception-v3 (Iv3$_{adv}$), Ens-Inception-Resnet-v2 (IRv2$_{ens}$) Tramèr et al. (2017), Adv-EfficientNet-b0 (Eb0$_{adv}$) to Adv-EfficientNet-b7 (Eb7$_{adv}$).

**Baselines.** For fair comparison of our EPN and conventional momentum, we replace conventional momentum with our EPN in momentum-based attacks, i.e., MI-FGSM (Dong et al., 2018), NI-

Table 3: The attack success rates (%) of adversarial examples crafted on source models against defense models. "Avg" means the average attack success rate.

| Models | Attacks | $Iv3_{adv}$ | $IRv2_{ens}$ | $Eb0_{adv}$ | $Eb1_{adv}$ | $Eb2_{adv}$ | $Eb3_{adv}$ | $Eb4_{adv}$ | $Eb5_{adv}$ | $Eb6_{adv}$ | $Eb7_{adv}$ | Avg |
|---|---|---|---|---|---|---|---|---|---|---|---|---|
| | MI-FGSM | 25.4 | 11.5 | 29.7 | 26.5 | 28.1 | 18.8 | 16.9 | 17.5 | 14.8 | 15.2 | 20.4 |
| | NI-FGSM | 25.4 | 11.6 | 34.3 | 31.0 | 31.2 | 22.2 | 18.6 | 18.6 | 16.0 | 16.5 | 22.5 |
| | EPNI-FGSM (**Ours**) | **32.4** | **14.3** | **49.0** | **45.0** | **45.4** | **31.1** | **23.9** | **25.7** | **22.4** | **22.3** | **31.2** |
| | DI-MI-FGSM | 33.0 | 19.3 | 46.6 | 42.9 | 42.4 | 31.3 | 27.4 | 26.7 | 22.9 | 24.4 | 31.7 |
| | TI-MI-FGSM | 29.9 | 21.5 | 35.2 | 32.6 | 34.9 | 27.8 | 29.3 | 28.2 | 24.5 | 26.6 | 29.1 |
| | SI-NI-FGSM | 37.4 | 23.2 | 50.2 | 46.1 | 46.3 | 33.3 | 30.9 | 29.6 | 27.6 | 25.8 | 35.0 |
| Iv3 | DI-EPNI-FGSM (**Ours**) | 41.5 | 25.2 | 65.0 | **65.5** | **64.7** | 45.5 | 39.1 | 41.6 | 36.7 | 35.6 | 46.0 |
| | TI-EPNI-FGSM (**Ours**) | **54.1** | **41.2** | 60.7 | 58.2 | 57.0 | 46.6 | **46.7** | **45.3** | 43.4 | **44.2** | **49.7** |
| | SI-EPNI-FGSM (**Ours**) | 47.4 | 28.4 | **66.3** | 64.7 | 63.1 | **46.7** | 41.8 | 42.1 | 37.9 | 37.3 | 47.6 |
| | VT-MI-FGSM | 36.8 | 25.1 | 50.5 | 46.3 | 44.9 | 33.5 | 29.4 | 29.4 | 26.8 | 26.2 | 34.9 |
| | VT-NI-FGSM | 39.1 | 26.4 | 54.0 | 50.6 | 49.5 | 36.6 | 31.0 | 32.7 | 30.3 | 28.5 | 37.9 |
| | VT-EPNI-FGSM (**Ours**) | **43.1** | **26.5** | **63.2** | **61.8** | **60.6** | **46.9** | **41.1** | **39.7** | **38.2** | **36.8** | **45.8** |
| | FI-MI-FGSM | 55.3 | 38.0 | 68.2 | 67.8 | 64.9 | 53.6 | 50.0 | 49.3 | 46.6 | 45.1 | 53.9 |
| | FI-EPNI-FGSM (**Ours**) | **58.9** | **39.9** | **76.2** | **73.3** | **72.7** | **61.1** | **56.0** | **55.6** | **52.9** | **50.5** | **59.7** |
| | MI-FGSM | 27.3 | 15.5 | 25.2 | 22.8 | 24.3 | 17.0 | 13.8 | 14.6 | 12.1 | 13.2 | 18.6 |
| | NI-FGSM | 25.9 | 14.9 | 24.8 | 23.1 | 23.6 | 17.8 | 14.1 | 14.9 | 12.8 | 13.7 | 18.6 |
| | EPNI-FGSM (**Ours**) | **33.5** | **15.9** | **36.3** | **32.4** | **34.9** | **23.9** | **20.7** | **20.4** | **17.6** | **17.8** | **25.3** |
| | DI-MI-FGSM | 33.1 | 24.7 | 38.1 | 35.8 | 37.9 | 27.1 | 24.6 | 23.2 | 22.0 | 20.8 | 28.7 |
| | TI-MI-FGSM | 38.1 | 32.2 | 39.0 | 36.7 | 39.4 | 30.2 | 33.8 | 31.4 | 29.2 | 29.7 | 34.0 |
| | SI-NI-FGSM | 37.0 | 33.0 | 50.8 | 48.2 | 46.9 | 35.7 | 32.0 | 28.7 | 26.9 | 27.5 | 36.7 |
| IRv2 | DI-EPNI-FGSM (**Ours**) | 39.4 | 29.7 | 57.4 | 52.4 | 53.4 | 41.0 | 33.6 | 33.3 | 30.2 | 30.6 | 40.1 |
| | TI-EPNI-FGSM (**Ours**) | **58.3** | **52.9** | 60.5 | 58.3 | 56.6 | 49.7 | **50.9** | **47.0** | **48.0** | **46.5** | **52.9** |
| | SI-EPNI-FGSM (**Ours**) | 45.6 | 41.5 | **68.9** | **64.3** | **66.8** | 50.2 | 42.6 | 43.5 | 39.7 | 40.1 | 50.3 |
| | VT-MI-FGSM | 38.8 | 36.4 | 42.3 | 38.9 | 41.2 | 31.2 | 28.4 | 26.4 | 25.4 | 25.4 | 33.4 |
| | VT-NI-FGSM | 40.5 | 34.8 | 44.6 | 41.1 | 43.4 | 33.1 | 28.2 | 26.6 | 25.8 | 26.2 | 34.4 |
| | VT-EPNI-FGSM (**Ours**) | **48.2** | **36.6** | **59.5** | **55.3** | **54.1** | **43.0** | **38.8** | **38.0** | **36.9** | **35.3** | **44.6** |
| | FI-MI-FGSM | 54.5 | 40.1 | 57.4 | 56.7 | 56.3 | 45.7 | 40.9 | 39.7 | 38.2 | 37.4 | 46.7 |
| | FI-EPNI-FGSM (**Ours**) | 53.3 | **47.1** | **59.6** | **56.8** | **57.9** | **47.0** | **43.3** | **42.1** | **40.6** | **38.6** | **48.6** |
| | MI-FGSM | 36.5 | 27.8 | 46.6 | 43.6 | 47.3 | 30.9 | 27.3 | 26.5 | 22.0 | 24.8 | 33.3 |
| | NI-FGSM | 40.1 | 29.4 | 49.8 | 47.3 | 47.8 | 33.2 | 29.1 | 27.7 | 25.2 | 25.4 | 35.5 |
| | EPNI-FGSM (**Ours**) | **53.1** | **43.1** | **71.9** | **68.1** | **70.5** | **50.2** | **41.0** | **41.0** | **37.5** | **40.2** | **51.7** |
| | DI-MI-FGSM | 57.6 | 51.7 | 72.6 | 68.3 | 71.2 | 54.7 | 47.1 | 44.7 | 44.2 | 44.3 | 55.6 |
| | TI-MI-FGSM | 46.8 | 41.1 | 50.7 | 46.4 | 50.1 | 41.8 | 43.2 | 41.0 | 36.2 | 38.7 | 43.6 |
| | SI-NI-FGSM | 51.7 | 43.4 | 62.8 | 57.3 | 61.2 | 43.4 | 39.3 | 36.0 | 35.4 | 33.8 | 46.4 |
| R152 | DI-EPNI-FGSM (**Ours**) | **78.3** | **73.1** | **91.6** | **89.9** | **90.6** | **75.7** | **67.3** | **64.9** | **62.6** | **64.3** | **75.8** |
| | TI-EPNI-FGSM (**Ours**) | 72.2 | 67.4 | 74.7 | 70.1 | 74.2 | 62.9 | 62.7 | 60.7 | 59.3 | 58.3 | 66.3 |
| | SI-EPNI-FGSM (**Ours**) | 67.5 | 59.3 | 79.4 | 78.5 | 81.1 | 61.8 | 53.7 | 51.5 | 48.8 | 50.3 | 63.2 |
| | VT-MI-FGSM | 59.1 | 52.6 | 69.6 | 64.9 | 69.3 | 55.1 | 48.8 | 46.1 | 43.6 | 44.6 | 55.4 |
| | VT-NI-FGSM | 61.6 | 55.5 | 73.6 | 68.2 | 72.1 | 56.1 | 50.6 | 48.7 | 45.5 | 47.9 | 58.0 |
| | VT-EPNI-FGSM (**Ours**) | **72.6** | **71.6** | **86.2** | **85.3** | **88.4** | **74.1** | **66.0** | **64.8** | **62.2** | **65.1** | **73.6** |
| | FI-MI-FGSM | 76.4 | 66.9 | 81.1 | 79.5 | 82.6 | 67.5 | 62.1 | 60.3 | 56.6 | 56.4 | 68.9 |
| | FI-EPNI-FGSM (**Ours**) | **81.0** | **73.3** | **88.5** | **87.0** | **88.6** | **76.3** | **70.9** | **70.1** | **66.4** | **66.4** | **76.9** |
| | MI-FGSM | 33.0 | 20.3 | 48.8 | 41.8 | 41.3 | 25.1 | 21.5 | 21.7 | 19.0 | 21.8 | 29.4 |
| | NI-FGSM | 33.0 | 22.5 | 49.5 | 43.6 | 44.3 | 27.7 | 22.2 | 22.7 | 18.9 | 20.1 | 30.5 |
| | EPNI-FGSM (**Ours**) | **46.7** | **29.0** | **71.2** | **65.7** | **63.5** | **41.5** | **31.3** | **31.5** | **28.5** | **28.7** | **43.8** |
| | DI-MI-FGSM | 44.9 | 31.6 | 62.4 | 55.9 | 56.9 | 37.5 | 32.0 | 30.9 | 27.8 | 28.1 | 40.8 |
| | TI-MI-FGSM | 38.9 | 29.4 | 44.3 | 38.0 | 41.4 | 31.1 | 32.0 | 31.5 | 28.5 | 28.2 | 34.3 |
| | SI-NI-FGSM | 50.0 | 33.5 | 61.4 | 56.4 | 54.4 | 36.5 | 29.9 | 30.0 | 27.2 | 26.4 | 40.6 |
| V16 | DI-EPNI-FGSM (**Ours**) | 61.2 | 44.7 | **80.3** | **77.9** | **78.2** | 52.2 | 43.5 | 42.6 | 40.2 | 38.8 | **56.0** |
| | TI-EPNI-FGSM (**Ours**) | 61.8 | **49.9** | 68.6 | 65.2 | 63.7 | 49.2 | **50.3** | **47.7** | **47.6** | **46.1** | 55.0 |
| | SI-EPNI-FGSM (**Ours**) | **62.8** | 42.6 | 78.6 | 75.4 | 74.3 | 48.1 | 40.6 | 41.9 | 37.9 | 35.5 | 53.8 |
| | VT-MI-FGSM | 49.4 | 35.4 | 65.3 | 58.7 | 58.6 | 39.7 | 34.3 | 33.6 | 29.2 | 32.2 | 43.6 |
| | VT-NI-FGSM | 49.3 | 37.5 | 67.3 | 60.2 | 60.9 | 40.7 | 35.1 | 34.0 | 29.8 | 32.0 | 44.7 |
| | VT-EPNI-FGSM (**Ours**) | **61.4** | **47.6** | **82.9** | **77.8** | **77.8** | **58.9** | **48.8** | **47.0** | **46.5** | **47.0** | **59.6** |
| | FI-MI-FGSM | 63.0 | 44.6 | 80.8 | 74.1 | 72.9 | 51.5 | 42.0 | 43.5 | 39.1 | 39.8 | 55.1 |
| | FI-EPNI-FGSM (**Ours**) | **64.7** | **44.8** | **85.1** | **81.3** | **80.8** | **58.6** | **46.2** | **49.1** | **42.5** | **42.4** | **59.6** |

FGSM (Lin et al., 2019), DI-MI-FGSM (Xie et al., 2019), TI-MI-FGSM (Dong et al., 2019), SI-NI-FGSM (Lin et al., 2019), VT-MI-FGSM (Wang & He, 2021), VT-NI-FGSM (Wang & He, 2021) and FI-MI-FGSM (Wang et al., 2021). Then we compare the transferability of conventional momentum-based attacks and our EPN-based attacks.

**Hyperparameters.** In all experiments, we follow the official default settings for hyperparameters. Specifically, the maximum perturbation $\epsilon = 16$, the number of iterations $T = 10$, the step size $\alpha = \epsilon/T = 1.6$, and the decay factor $\mu = 1.0$. For DIM (Xie et al., 2019), the probability $p$ is set to 0.5. For TIM (Dong et al., 2019), the size of the Gaussian kernel is set to 15×15. For SIM (Lin et al., 2019), the number of scale copies $m$ is set to 5. For VT-MI-FGSM (Wang & He, 2021) and VT-NI-FGSM (Wang & He, 2021), the number of sampled examples $N$ is set to 20, and the parameter $\beta$ for the upper bound of the neighborhood is set to 1.5. For FI-MI-FGSM (Wang et al., 2021), the drop

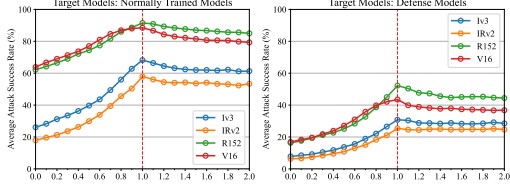 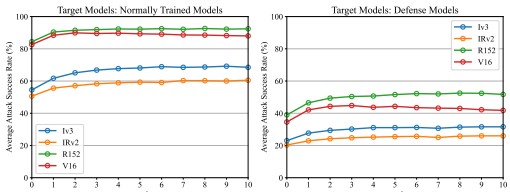

Figure 3: The average attack success rates (%) of the adversarial examples crafted on source models against normally trained models (except the source model) and defense models for various $\mu$.

Figure 4: The average attack success rates (%) of the adversarial examples crafted on source models against normally trained models (except the source model) and defense models for various $epochs$.

probability $p_d$ is set to 0.3 when attacking normally trained models and 0.1 when attacking defense models, the ensemble number $N$ is set to 30 in aggregate gradient, and the intermediate layer is set to *Mixed_5b* for Iv3, *Conv_4a* for IRv2, *Conv3_3* for V16 as well as the last layer of the second block for R152. For our EM-based attacks, $epochs$ is set to 5.

## 4.2 ATTACK NORMALLY TRAINED MODELS

To validate that EPN-based attacks have higher transferability than conventional momentum-based attacks, we choose Iv3, IRv2, R152, and V16 as the source model, respectively, and attack normally trained target models via our EPN-based methods and baseline methods. The attack success rates are shown in Table 2. The results show that the attack success rates of our EPN-based methods are $\sim$11.9% higher than baseline methods on average, In particular, our EPN-based attacks have the best transferability against normally trained target models when the source model is R152. Specifically, the attack success rates of EPNI-FGSM, DI-EPNI-FGSM, VT-EPNI-FGSM, and FI-EPNI-FGSM are 92.0%, 97.2%, 96.0%, and 96.0%, respectively, on average. Therefore, the experiments demonstrate that our EPN improves transferability more effectively than conventional momentum against normally trained models.

## 4.3 ATTACK DEFENSE MODELS

To further compare the transferability, we also use defense models as the target models, and the source models are still Iv3, IRv2, R152, and V16. We craft adversarial examples on the source model via our EPN-based methods and baseline methods to attack defense models. The attack success rates are shown in Table 3. The results show that the attack success rates of our EPN-based methods are $\sim$13.1% higher than baseline methods on average. Adversarial examples crafted on R152 still show the best transferability against defense models. Specifically, the attack success rates of EPNI-FGSM, DI-EPNI-FGSM, VT-EPNI-FGSM, and FI-EPNI-FGSM are 51.7%, 75.8%, 73.6%, and 76.9%, respectively, on average. The results of experiments indicate that our EPN is still more effective than conventional momentum against defense models.

## 4.4 ABLATION STUDY

We conduct ablation studies for EPNI-FGSM. We investigate the impacts of two hyperparameters (i.e., the decay factor $\mu$ and the epochs of pretraining $epochs$) on the transferability of EPNI-FGSM in Sec. 4.4.1. We further study the impacts of EM and PN on transferability in Sec. 4.4.2.

### 4.4.1 IMPACTS OF $\mu$ AND $epochs$

The source models are set to Iv3, IRv2, R152, and V16. We use EPNI-FGSM to craft adversarial examples to attack normally trained models and defense models, respectively. We investigate the impacts of $\mu$ and $epochs$ on the transferability of EPNI-FGSM by counting the average attack success rates against normally trained models (except the source model) and defense models.

**The decay factor $\mu$.** The decay factor $\mu$ plays a vital role for momentum. If $\mu = 0$, the momentum-based attacks degrade to vanilla iterative gradient-based attacks. If $0 < \mu < 1$, the previous gradients

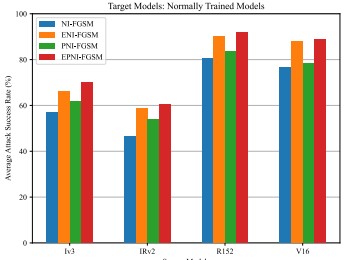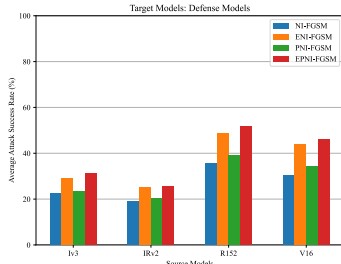

Figure 5: The average attack success rates (%) of the adversarial examples crafted on source models against normally trained models and defense models via NI-FGSM, ENI-FGSM, PNI-FGSM, and EPNI-FGSM.

accumulated in the momentum decay exponentially. If $\mu = 1$, the momentum simply adds up all previous gradients. If $\mu > 1$, the previous gradients accumulated in the momentum grow exponentially. We pre-set $epochs = 5$ and set $\mu$ from 0.0 to 2.0 with a step size of 0.1. The average attack success rates are shown in Fig. 3. When $\mu \leq 1.0$, the average attack success rates show an upward trend, and when $\mu \geq 1.0$, the average attack success rates show a downward trend. Therefore, we set $\mu = 1.0$ for EPNI-FGSM to achieve the best transferability.

**The epochs of pretraining** $epochs$. $epochs$ affects the amount of gradient accumulated in EM. We pre-set $\mu = 1.0$ and set $epochs$ from 0 to 10 with a step size of 1. The average attack success rates are shown in Fig. 4. As $epochs$ increases, the average attack success rates increase and gradually converge. Since the larger $epochs$, the higher the computational cost, we set $epochs = 5$ for EPNI-FGSM to strike a balance between computational cost and transferability.

In summary, we set the decay factor $\mu = 1.0$ and the epochs of pretraining $epochs = 5$ for EPNI-FGSM. Similarly, such two hyperparameters of other EPN-based attacks have the same settings as EPNI-FGSM.

### 4.4.2 IMPACTS OF EM AND PN

The source models are the same as in Sec 4.4.1. To investigate the impacts of EM and PN, we craft adversarial examples on source models via ENI-FGSM (only with EM), PNI-FGSM (only with PN), and EPNI-FGSM (with EM and PN), respectively. In addition, we also use MI-FGSM and NI-FGSM (without EM and PN) as baselines. For ENI-FGSM, the epochs of pretraining $epochs$ is set to 5. For ENI-FGSM and PNI-FGSM, the decay factor $\mu$ is set to 1.0. The average attack success rates of the adversarial examples against normally trained models and defense models are shown in Fig. 5. The average attack success rates of ENI-FGSM are higher than MI-FGSM and NI-FGSM, demonstrating that EM improves transferability more than conventional momentum. The same is true for PN. Besides, the average attack success rates of EPNI-FGSM are higher than that of ENI-FGSM and PNI-FGSM, demonstrating that the combination of EM and PN can further improve transferability.

## 5 CONCLUSION

In this work, we proposed Experienced Momentum (EM) and Precise Nesterov momentum (PN) to boost transferability. Specifically, EM is trained on a set of derived models by Random Channels Swapping (RCS), and then conventional momentum is initialized to EM, which can accelerate optimization to escape from saddle points and poor local extrema during early iterations to improve transferability. Additionally, we adopted the current gradient to refine the pre-update of conventional Nesterov momentum, called PN. Then, we naturally combined EM and PN as EPN to improve transferability further. Extensive experiments demonstrate that EPN-based attacks have higher transferability than conventional momentum-based attacks. However, our methods still adopt a fixed learning rate or step size that is crucial for the optimizer. Therefore, we will explore the impact of learning rate or step size on transferability in future work.

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
