# OpenReview forum: "Improving the Transferability of Adversarial Attacks through Experienced Precise Nesterov Momentum"
_ICLR.cc/2023/Conference — Submitted to ICLR 2023_

### Official Review · Reviewer_shJD · 2022-10-17

**Confidence:** 5
**Correctness:** 2
**Technical Novelty And Significance:** 2
**Empirical Novelty And Significance:** 1
**Recommendation:** 3

**Clarity, Quality, Novelty And Reproducibility:**

The expression of this paper is relatively clear but the novelty is limited. It should be able to reproduce well.

**Strength And Weaknesses:**

Strengths:
1. EPN focuses on better optimization methods and random channel data enhancement to improve transferability.
2. This work proposes Experienced Momentum (EM) with Random Channels Swapping (RCS) for accelerating optimization effectively.
3. This work proposes Precise Nesterov momentum (PN) to complete a more precise pre-attack to help momentum out of the local optimal solution.

Weaknesses:
1. The overall motivational analysis part of the paper is lacking. The authors do not adequately explain the motivation for the pre-trained momentum and the PN. Furthermore, there is also no relevant citations or empirical experiments as support.
2. The experiments are done only for some basic models and defense models, and do not consider the performance of attacks under ensemble models and advanced defense methods (HGD, R&P, NIPS-R3, etc.), resulting in a lack of experimental completeness. In model selection, homologous models are simply stacked too much. Ensemble models and advanced defense methods are necessary for previous work on transfer attacks [1][2][3].
3. The benchmark method uses for the experiments do not consider methods with better performance, such as VT-SITIDI-MIFGSM [3]. Only a relatively single VT method is considered, and the performance in the paper does not exceed the current SOTA method [3].
4. The pre-trained momentum introduces additional attack iterations. Can a more iterative attack with a common method achieve a similar effect?

[1] Dong, Yinpeng, Tianyu Pang, Hang Su, and Jun Zhu. "Evading defenses to transferable adversarial examples by translation-invariant attacks." In Proceedings of the IEEE/CVF Conference on Computer Vision and Pattern Recognition, pp. 4312-4321. 2019.
[2] Gao, Lianli, Qilong Zhang, Jingkuan Song, Xianglong Liu, and Heng Tao Shen. "Patch-wise attack for fooling deep neural network." In European Conference on Computer Vision, pp. 307-322. Springer, Cham, 2020.
[3] Wang, Xiaosen, and Kun He. "Enhancing the transferability of adversarial attacks through variance tuning." In Proceedings of the IEEE/CVF Conference on Computer Vision and Pattern Recognition, pp. 1924-1933. 2021.


**Summary Of The Paper:**

This paper focuses on the transferability of black-box adversarial attacks. As a benchmark method, the introduction of momentum in iterative attacks helps the attack to get out of the local optimal solution for better transfer attack performance. The authors argue that the initial setting of zero in momentum inhibits the effect of momentum so propose Experienced Momentum (EM) with Random Channels Swapping (RCS) for accelerating optimization effectively. On the other hand, for problems of imprecise pre-update of the traditional Nesterov momentum, authors propose Precise Nesterov momentum (PN), which pre-updates by considering the gradient of the current data point. Finally, combining the above two methods to obtain EPN and achieve state-of-the-art results on some methods and models.

**Summary Of The Review:**

We tend to reject this paper due to lack of motivational analysis and necessary experiments.

---

> ### Author Response · Authors · 2022-11-13
> **motivation, ensemble in logits, and advanced defenses**
>
> We thank reviewer ${\color{Green} shJD}$ for the valuable reviews.
>
> ---
>
> **Review1:** The overall motivational analysis part of the paper is lacking.
>
> **Response1:** This effective review will qualitatively improve the paper.
> We are adding motivational analysis to the paper under revision.
>
> ---
>
> **Review2:** The experiments do not consider the performance of attacks under ensemble models and advanced defense methods.
>
> **Response2:** We considered crafting adversarial examples on ensemble models and
> testing transferability on hold-out models (called *ensemble in logits* [1]). As an ensemble method,
> ensemble in logits can theoretically improve the transferability of our attacks, and we have experimentally verified this assertion.
> In this case, our attacks are more transferable than baselines attacks, and
> ensemble in logits + our attacks are also more transferable than ensemble in logits + baselines.
> Therefore, we choose the most basic baselines for comparison to demonstrate more concisely that our EPN is better than the conventional momentum.
> However, we are willing to show more experiments in the paper under revision.
> Some experimental results are as follows:
>
> | Models | Attacks                     | Inc-v3     | Inc-v4   | IncRes-v2 | Res-50    | Res-152    | Vgg-16     | Vgg-19    |
> |--------|-----------------------------|------------|----------|-----------|-----------|------------|------------|-----------|
> | Inc-v3 | VT-DTS-MI-FGSM              | 99.3*      | 73.7     | 68.7      | 74.8      | 70.5       | 77.0       | 77.6      |
> |        | VT-DTS-EPNI-FGSM (**Ours**) | **100.0*** | **88.1** | **83.1**  | **87.0**  | **84.8**   | **86.5**   | **89.4**  |
> | Ens    | VT-DTS-MI-FGSM              | **100.0*** | 95.9     | 92.8*     | **100.0** | **100.0*** | **100.0*** | **100.0** |
> |        | VT-DTS-EPNI-FGSM (**Ours**) | **100.0*** | **99.4** | **97.1*** | **100.0** | **100.0*** | **100.0*** | **100.0** |
>
> | Models | Attacks                     | Adv-Inc-v3 | Ens-IncRes-v2 | R&P      | Bit-Red  | JPEG     | FD       |
> |--------|-----------------------------|------------|---------------|----------|----------|----------|----------|
> | Inc-v3 | VT-DTS-MI-FGSM              | 69.7       | 57.7          | 55.9     | 59.3     | 66.6     | 60.8     |
> |        | VT-DTS-EPNI-FGSM (**Ours**) | **79.3**   | **71.8**      | **74.2** | **74.5** | **82.4** | **77.9** |
> | Ens    | VT-DTS-MI-FGSM              | 91.2       | 86.3          | 83.4     | 87.9     | 90.6     | 88.2     |
> |        | VT-DTS-EPNI-FGSM (**Ours**) | **93.9**   | **89.5**      | **87.1** | **90.9** | **92.6** | **91.3** |
>
> where DTS denotes the ensemble of DIM [3], TIM [4], and SIM [2]. We report the results for an ensemble of
> white-box models and a hold-out black-box target model.
> The experimental results show that our EPN also shows better performance in improvement of transferability under ensemble model setting.
>
> Sorry that we have not been able to successfully reproduce some defenses (e.g., HGD and NIPS-r3), since the pytorch version of the code is not officially released
>
> ---
>
> **Review3:** The benchmark method uses for the experiments do not consider methods with better performance, such as VT-SITIDI-MIFGSM.
>
> **Response3:** In this paper, we have made the following comparisons to demonstrate that our EPN is better than the
> conventional momentum (Polyak and Nesterov momentum) in improvement of transferability:
>
> - Polyak or Nesterov momentum based attack [1, 2] *vs.* EPN-based attack
> - Polyak or Nesterov momentum based attack + input transformations [3, 4, 2] *vs.* EPN-based attack + input transformations
> - Polyak or Nesterov momentum based attack + variance tuning [5] *vs.* EPN-based attack + variance tuning
> - Polyak or Nesterov momentum based attack + feature importance-aware [6] *vs.* EPN-based attack + feature importance-aware
>
> We considered more comparisons of ensemble attacks (e.g., Polyak or Nesterov momentum based attack + variance tuning + input transformations *vs.* EPN-based attack + variance tuning + input transformations).
> Theoretically, our method remains advanced in the same combination of cases, and we have experimentally verified this assertion (the previous table).
> In this case, we chose to compare in the most basic case to exclude any potential bugs.
> However, we are willing to show more experiments in the paper under revision.

---

> > ### Comment · Reviewer_shJD · 2022-12-01
> > **Acknowledgement of Rebuttal**
> >
> > I would like to thank the authors for the detailed feedback and additional results.
> >
> > After reading the response, I still feel that the paper is not ready for publication.
> > First, the motivation of  the paper is still unconvinced. Second, it is not rigorous to not do comparison experiments just because there is no torch version of the source code. I keep my score.

---

> > > ### Author Response · Authors · 2022-12-09
> > > **About Motivation**
> > >
> > > Given an input image $\boldsymbol{x} \in \mathbb{R}^n$, a white-box source model $f(\boldsymbol{x}; \boldsymbol{\theta}): \mathbb{R}^n \to \mathbb{R}^l$ with accessible parameters $\boldsymbol{\theta}$, and a black-box target model $f'(\boldsymbol{x}; \boldsymbol{\theta}'): \mathbb{R}^n \to \mathbb{R}^l$ with inaccessible parameters $\boldsymbol{\theta}'$. Assume that $f$ owns a loss function $\mathcal{J}(\boldsymbol{x},y): \mathbb{R}^n \times \{1, \cdots, l\} \to \mathbb{R}^+$ (e.g., the cross-entropy loss), where $y \in \{1, \cdots, l\}$ is the ground-truth label of $\boldsymbol{x}$. Non-targeted transfer-based attacks can be described as the following two steps: (i) crafting the adversarial example $\boldsymbol{x}^{adv}$ on the source model $f$ by maximizing loss, i.e.,
> > >
> > >
> > > \begin{eqnarray}
> > > \underset{\boldsymbol{x}^{adv}}{\arg \max} \mathcal{J}(\boldsymbol{x}^{adv}, y), \quad \mathrm{s.t.} || \boldsymbol{x}^{adv} - \boldsymbol{x} ||_p \le \epsilon,~~~~(1)
> > > \end{eqnarray}
> > >
> > >
> > > where $\epsilon$ denotes the maximum perturbation, $|| \cdot ||_p$ denotes the $L_p$ norm, and $p$ is generally $0$, $1$, $2$, $\infty$; (ii) transferring the adversarial example $\boldsymbol{x}^{adv}$ to the target model $f'$ to satisfy $f'(\boldsymbol{x}; \boldsymbol{\theta}') \ne f'(\boldsymbol{x}^{adv}; \boldsymbol{\theta}')$. Note that we focus on non-targeted attacks to align with previous works and our proposed methods can be easily transferred to targeted attacks by replacing the objective function in Eq. 1 with $-\mathcal{J}(\boldsymbol{x}^{adv}, y^*)$, where $y^*$ denotes the target label.
> > >
> > > The adversarial example can easily drop into poor local maxima and “overfit” the source model, leading to being less likely to transfer the adversarial example to the target model. Generally, momentum can stabilize the updates of the adversarial example and help the adversarial example to escape from poor local maxima. Thus, integration with momentum to improve transferability has become the basis for previous transfer-based attacks.
> > >
> > > However, momentum is invariably initialized to zero, resulting in the inefficiency of momentum during early iterations. Therefore, we consider assigning an initial value to momentum in order to stabilize the updates of the adversarial example easier during early iterations and make the adversarial example escape from poor local maxima faster. Furthermore, the pre-update of the Nesterov momentum based attack is probably rough. Specifically, Nesterov momentum based attacks pre-update along momentum to obtain the pre-update point, which is an estimate of the next position. Nevertheless, the pre-update only takes into consideration the previous gradients and ignores the information of the current data point, making the estimate of the next position likely to be imprecise. Hence, we attempted to consider the information of the current data point during the pre-update to get more precise estimation of the next position. Based on the above analysis, we propose EM and PN to improve transferability as well as combining EM and PN as EPN to obtain stronger transferability.

---

> > > ### Author Response · Authors · 2022-12-09
> > > **About Reproducing**
> > >
> > > We would like to thank the reviewer for the rebuttal.
> > >
> > > ---
> > >
> > > In the case of no PyTorch version of the source code, it is truly not rigorous to insist on reproducing it, which is likely to lead to experimental results that are imperceptibly wrong or not robust.

---

> ### Author Response · Authors · 2022-11-13
> **About additional attack iterations**
>
> We thank reviewer ${\color{Green} shJD}$ for the constructive review.
>
> ---
>
> **Review4:** The pre-trained momentum introduces additional attack iterations.
>
> **Response4:** Thanks for your valuable review. We are sorry that we have not made it clear enough in our initial submission.
> The iteration steps $T$ for adversarial examples is independent of steps 2 to 9 in Algorithm 1.
>
> - Firstly, the original image $x$ is **pre-perturbed** in steps 2 to 9.
> - Secondly, the algorithm conducts a critical operation in step 10: $x^{adv}_1 \gets x$. This operation initializes the adversarial example $x^{adv}_1$ to the original image $x$.
> - Thirdly, the algorithm only updates the adversarial example $x^{adv}_t$ $T$ times in steps 11 to 13
> (the iteration steps of all baselines are also set to T).
> For each iteration, the direction is the sign of the momentum, and the step size is $alpha$.
>
> In general, the effect of steps 2 to 9 is to give the momentum an initial value ($g_0 \gets g^{exp}$, in step 10),
> which gives the momentum more iterations, but does not affect the iteration steps for the adversarial sample.
>
> ---
>
> [1] Yinpeng Dong, Fangzhou Liao, Tianyu Pang, Hang Su, Jun Zhu, Xiaolin Hu, and Jianguo Li. Boosting adversarial attacks with momentum. In Proceedings of the IEEE conference on computer
> vision and pattern recognition, pp. 9185–9193, 2018.
>
> [2] Jiadong Lin, Chuanbiao Song, Kun He, Liwei Wang, and John E Hopcroft. Nesterov accelerated
> gradient and scale invariance for adversarial attacks. arXiv preprint arXiv:1908.06281, 2019.
>
> [3] Cihang Xie, Zhishuai Zhang, Yuyin Zhou, Song Bai, Jianyu Wang, Zhou Ren, and Alan L Yuille.
> Improving transferability of adversarial examples with input diversity. In Proceedings of the
> IEEE/CVF Conference on Computer Vision and Pattern Recognition, pp. 2730–2739, 2019.
>
> [4] Yinpeng Dong, Tianyu Pang, Hang Su, and Jun Zhu. Evading defenses to transferable adversarial examples by translation-invariant attacks. In Proceedings of the IEEE/CVF Conference on
> Computer Vision and Pattern Recognition, pp. 4312–4321, 2019.
>
> [5] Xiaosen Wang and Kun He. Enhancing the transferability of adversarial attacks through variance
> tuning. In Proceedings of the IEEE/CVF Conference on Computer Vision and Pattern Recognition, pp. 1924–1933, 2021.
>
> [6] Zhibo Wang, Hengchang Guo, Zhifei Zhang, Wenxin Liu, Zhan Qin, and Kui Ren. Feature
> importance-aware transferable adversarial attacks. In Proceedings of the IEEE/CVF International
> Conference on Computer Vision, pp. 7639–7648, 2021.

---

### Official Review · Reviewer_u7X7 · 2022-10-24

**Confidence:** 4
**Correctness:** 3
**Technical Novelty And Significance:** 2
**Empirical Novelty And Significance:** Not applicable
**Recommendation:** 5

**Clarity, Quality, Novelty And Reproducibility:**

Clarity: The writing is clear and the paper is easy to follow.
Quality: The experimental results are impressive, but lack of strong theoretical support.
Novelty: An increment on existing methods.
Reproducibility: Not applicable.


**Strength And Weaknesses:**

Strengths:
-	Extensive experiments show that EPN significantly improves the transferability of various momentum based adversarial attacks.
Weaknesses:
-	The approach is not tested on ensembles of models. Neither does it be tested on advanced defense models other than AT, such as Randomized Smoothing (RS), Neural Representation Purifier (NRP).
-	Ablation experiments to verify the effectiveness of RCS are missing in the paper.
-	EM requires a few iterations first. NI doubles the time it takes to compute the gradient. The efficiency of the algorithm needs to be discussed more in this paper.


**Summary Of The Paper:**

This paper proposes two methods, Experienced Momentum (EM) and Precise Nesterov momentum (PN), to improve the transferability of adversarial attacks. Specifically, the proposed method starts with a few iterations to accumulate the gradient, and uses it as the initial momentum. PN considers the gradient of the current data point in the pre-update of Nesterov momentum to make the pre-update precise. The combination of EM and PN boosts the transferability of conventional momentum based adversarial attacks.

**Summary Of The Review:**

The experimental results are impressive, but lack of strong theoretical support.

---

> ### Author Response · Authors · 2022-11-13
> **ensemble in logits and advanced defense models**
>
> We thank reviewer ${\color{Red} u7X7}$ for the detailed review.
>
> ---
>
> **Review1:** The approach is not tested on ensembles of models.
> Neither does it be tested on advanced defense models other than AT.
>
> **Response1:** We considered crafting adversarial examples on ensemble models and
> testing transferability on hold-out models (called *ensemble in logits* [1]). As an ensemble method,
> ensemble in logits can theoretically improve the transferability of our attacks, and we have experimentally verified this assertion.
> In this case, our attacks are more transferable than baselines attacks, and
> ensemble in logits + our attacks are also more transferable than ensemble in logits + baselines.
> Therefore, we choose the most basic baselines for comparison to demonstrate more concisely that our EPN is better than the conventional momentum.
> However, we are willing to show more experiments in the paper under revision.
> Some experimental results are as follows:
>
> | Models | Attacks                     | Inc-v3     | Inc-v4   | IncRes-v2 | Res-50    | Res-152    | Vgg-16     | Vgg-19    |
> |--------|-----------------------------|------------|----------|-----------|-----------|------------|------------|-----------|
> | Inc-v3 | VT-DTS-MI-FGSM              | 99.3*      | 73.7     | 68.7      | 74.8      | 70.5       | 77.0       | 77.6      |
> |        | VT-DTS-EPNI-FGSM (**Ours**) | **100.0*** | **88.1** | **83.1**  | **87.0**  | **84.8**   | **86.5**   | **89.4**  |
> | Ens    | VT-DTS-MI-FGSM              | **100.0*** | 95.9     | 92.8*     | **100.0** | **100.0*** | **100.0*** | **100.0** |
> |        | VT-DTS-EPNI-FGSM (**Ours**) | **100.0*** | **99.4** | **97.1*** | **100.0** | **100.0*** | **100.0*** | **100.0** |
>
> | Models | Attacks                     | Adv-Inc-v3 | Ens-IncRes-v2 | R&P      | Bit-Red  | JPEG     | FD       |
> |--------|-----------------------------|------------|---------------|----------|----------|----------|----------|
> | Inc-v3 | VT-DTS-MI-FGSM              | 69.7       | 57.7          | 55.9     | 59.3     | 66.6     | 60.8     |
> |        | VT-DTS-EPNI-FGSM (**Ours**) | **79.3**   | **71.8**      | **74.2** | **74.5** | **82.4** | **77.9** |
> | Ens    | VT-DTS-MI-FGSM              | 91.2       | 86.3          | 83.4     | 87.9     | 90.6     | 88.2     |
> |        | VT-DTS-EPNI-FGSM (**Ours**) | **93.9**   | **89.5**      | **87.1** | **90.9** | **92.6** | **91.3** |
>
> where DTS denotes the ensemble of DIM [3], TIM [4], and SIM [2]. We report the results for an ensemble of
> white-box models and a hold-out black-box target model.
> The experimental results show that our EPN also shows better performance in improvement of transferability under ensemble model setting.
>
> Sorry that we have not been able to successfully reproduce some defenses (e.g., RS and NRP), since the pytorch version of the code is not officially released.
>
> ---
>
> [1] Yinpeng Dong, Fangzhou Liao, Tianyu Pang, Hang Su, Jun Zhu, Xiaolin Hu, and Jianguo Li. Boosting adversarial attacks with momentum. In Proceedings of the IEEE conference on computer
> vision and pattern recognition, pp. 9185–9193, 2018.
>
> [2] Jiadong Lin, Chuanbiao Song, Kun He, Liwei Wang, and John E Hopcroft. Nesterov accelerated
> gradient and scale invariance for adversarial attacks. arXiv preprint arXiv:1908.06281, 2019.
>
> [3] Cihang Xie, Zhishuai Zhang, Yuyin Zhou, Song Bai, Jianyu Wang, Zhou Ren, and Alan L Yuille.
> Improving transferability of adversarial examples with input diversity. In Proceedings of the
> IEEE/CVF Conference on Computer Vision and Pattern Recognition, pp. 2730–2739, 2019.
>
> [4] Yinpeng Dong, Tianyu Pang, Hang Su, and Jun Zhu. Evading defenses to transferable adversarial examples by translation-invariant attacks. In Proceedings of the IEEE/CVF Conference on
> Computer Vision and Pattern Recognition, pp. 4312–4321, 2019.

---

> > ### Comment · Reviewer_u7X7 · 2022-12-13
> > **Acknowledgement of Rebuttal**
> >
> > Thanks for the detailed reply. The reply addresses my concern about the ablation experiments of RCS. However, for testing on ensembles of models and advanced defense models other than AT, I checked [1] and found that the results of VT-DTS-MI-FGSM in the rebuttal is much lower than that reported in [1]. I think the experiment should be checked and complemented before publication. I keep my score.
> >
> > [1] Xiaosen Wang, Kun He. Enhancing the Transferability of Adversarial Attacks Through Variance Tuning. CVPR 2021

---

> ### Author Response · Authors · 2022-11-13
> **Ablation for RCS and the efficiency of the algorithm**
>
> We thank reviewer ${\color{Red} u7X7}$ for the thorough review on our work.
>
> ---
>
> **Review2:** Ablation experiments to verify the effectiveness of RCS are missing in the paper.
>
> **Response2:** Sorry that we lacked rigor. We are completing the ablation study of RCS, and some of the results are shown as follows:
>
> | Models | Attacks                | Inc-v3    | Inc-v4   | IncRes-v2 | Res-50   | Res-152  | Vgg-16   | Vgg-19   |
> |--------|------------------------|-----------|----------|-----------|----------|----------|----------|----------|
> | Inc-v3 | MI-FGSM                | **100.0** | 49.6     | 42.5      | 46.4     | 39.9     | 53.9     | 52.1     |
> |        | EMI-FGSM (without RCS) | **100.0** | 52.3     | 46.5      | 47.7     | 41.6     | 58.6     | 59.6     |
> |        | EMI-FGSM (with RCS)    | **100.0** | **65.0** | **58.0**  | **51.6** | **45.3** | **63.5** | **63.6** |
>
> | Models | Attacks                | Adv-Inc-v3 | Ens-IncRes-v2 | R&P      | Bit-Red  | JPEG     | FD       |
> |--------|------------------------|------------|---------------|----------|----------|----------|----------|
> | Inc-v3 | MI-FGSM                | 25.4       | 11.5          | 11.3     | 13.7     | 21.8     | 20.0     |
> |        | EMI-FGSM (without RCS) | 28.7       | 11.9          | 11.6     | 14.5     | 24.1     | 22.9     |
> |        | EMI-FGSM (with RCS)    | **31.7**   | **12.5**      | **12.2** | **16.8** | **27.4** | **25.3** |
>
> Experimental results show that RCS is useful to improve the transferability of EM-based attacks.
>
> ---
>
> **Review3:** The efficiency of the algorithm needs to be discussed more in this paper.
>
> **Response3:** Our EPN does have a higher time cost than conventional momentum, but it is still acceptable.
> The time complexity of the attacks and the running time for each adversarial sample are shown as follows:
>
> | Models | Attacks   | Time complexity   | Times (s) |
> |--------|-----------|-------------------|-----------|
> | Inc-v3 | MI-FGSM   | $O(T)$              | 0.108     |
> |        | NI-FGSM   | $O(T)$              | 0.108     |
> |        | EMI-FGSM  | $O(epochs * T)$     | 0.582     |
> |        | ENI-FGSM  | $O(epochs * T)$     | 0.582     |
> |        | PNI-FGSM  | $O(2 * T)$          | 0.196     |
> |        | EPNI-FGSM | $O(2 * epochs * T)$ | 1.136     |
>
> The results show that time complexity and the running time basically match,
> and the running time of EPNI-FGSM is about $2 * epochs$ times higher than MI-FGSM and NI-FGSM, but it is still acceptable.

---

### Official Review · Reviewer_aDJy · 2022-10-24

**Confidence:** 4
**Correctness:** 2
**Technical Novelty And Significance:** 3
**Empirical Novelty And Significance:** 3
**Recommendation:** 6

**Clarity, Quality, Novelty And Reproducibility:**

- the paper is clearly written but it is suggested to give more explanation on the method: why it is more accurate and why accurate second order information helps adversarial transferbility.

- the method looks novel but I do not know whether there is similar idea in training (not adversarial attack) and optimization, for which there are many modification on momentum.



**Strength And Weaknesses:**

+ the performance in experiments are quite good compared to existing  MI-FGSM.

- the necessity of predicting of accurate momentum is lacking. Accurate momentum is helping optimization, but why it can lead better adversarial transferbility?

- the proposed method is claimed to have advantages over standard  Polyak momentum and Nestrov momentum, then I am wondering how about the performance on DNN training tasks?

- momentum is an estimation for second-order information, without comparing with Hessian, it is hard to say which estimation is more accurate, i.e., I do not like the title saying precise momentum.

- following the idea that "accurate" momentum can help, then can we expect that some second-order method can further improve the transferbility?

**Summary Of The Paper:**

In this paper, the authors modify the momentum term by two novel methods to improve transferability of adversarial examples.

**Summary Of The Review:**

The performance of this work is quite promising. The main weakness is the lacking of explanation and at least I am not convinced why those modification can help adversarial transferbility. Thus, I will give a borderline score and am willing to change it during rebuttal.

---

> ### Author Response · Authors · 2022-11-10
> **Explanation, the performance on DNN training tasks, and prospect**
>
> **Review1:** Accurate momentum is helping optimization, but why it can lead better adversarial transferability?
>
> **Response1:** The poor transferability of gradient-based attacks can be explained as that the adversarial example can
> easily drop into poor local maxima and "overfit" the model, which is not likely to transfer across models [1].
> The momentum can stabilize the optimization and help the adversarial sample to escape from poor
> local maxima, thus improving the transferability of the adversarial sample.
> Within limited iterations, more accurate momentum can help the adversarial sample to escape
> from poor local maxima easier and faster [2], leading to stronger transferability.
>
> ---
>
> **Review2:** I am wondering how about the performance on DNN training tasks?
>
> **Response2:** The classifiers are all pre-trained models, and they are obtained from torchvision, timm, and efficientnet_pytorch.
> The accuracy of classifiers on the DEV dataset from the NIPS17 Adversarial Attacks and Defenses Competition is as follows:
>
> | Iv1  | Iv3  | Iv4  | IRv2 | R18  | R34  | R50  | R101 | R152 | V11  | V13  | V16  | V19  | D121 | D169 | D201 | D161 |
> |------|------|------|------|------|------|------|------|------|------|------|------|------|------|------|------|------|
> | 91.2 | 95.1 | 94.8 | 97.2 | 84.3 | 88.7 | 92.8 | 92.4 | 94.5 | 82.3 | 83.3 | 86.6 | 89.1 | 92.5 | 94.5 | 93.7 | 95.2 |
>
> | Adv-Iv3 | Ens-IRv2 | Adv-Eb0 | Adv-Eb1 | Adv-Eb2 | Adv-Eb3 | Adv-Eb4 | Adv-Eb5 | Adv-Eb6 | Adv-Eb7 |
> |---------|----------|---------|---------|---------|---------|---------|---------|---------|---------|
> | 87.0    | 94.5     | 93.0    | 94.4    | 94.9    | 93.8    | 92.3    | 90.9    | 93.2    | 95.0    |
>
> ---
>
> **Review3:** momentum is an estimation for second-order information, without comparing with Hessian, it is hard to say which estimation is more accurate.
>
> **Response3:** The pre-update of Precise Nesterov momentum (PN) is more detailed than Nesterov momentum,
> and PN-based attacks are more transferable than Nesterov momentum based attacks,
> so we use the term "precise" to describe PN.
>
> ---
>
> **Review4:** can we expect that some second-order method can further improve the transferability?
>
> **Response4:** That sounds great, thanks for the suggestion. We will answer this question in our future work.
>
> ---
>
> [1] Yinpeng Dong, Fangzhou Liao, Tianyu Pang, Hang Su, Jun Zhu, Xiaolin Hu, and Jianguo Li. Boosting adversarial attacks with momentum. In Proceedings of the IEEE conference on computer
> vision and pattern recognition, pp. 9185–9193, 2018.
>
> [2] Jiadong Lin, Chuanbiao Song, Kun He, Liwei Wang, and John E Hopcroft. Nesterov accelerated
> gradient and scale invariance for adversarial attacks. arXiv preprint arXiv:1908.06281, 2019. <br>

---

> > ### Comment · Reviewer_aDJy · 2022-11-21
> > **thanks for the reply**
> >
> > Thanks for the reply and discussion.
> >
> > For my question "Accurate momentum is helping optimization, but why it can lead better adversarial transferability?". I certainly know the previous paper but I want to get the authors' opinion about this, not only empirical observations. But it is OK, if the authors do not have new findings. If there is, I am looking forwards to that.
> >
> > For my question "I am wondering how about the performance on DNN training tasks?", which is not to ask the performance of the pre-trained model. This is following the first question. If your answer  "more accurate momentum can help the adversarial sample to escape from poor local maxima easier and faster" is true, then I expect that the proposed method is also helpful for regular training task. That is, can the proposed method could generalize better than SGD or Adam? If yes, I would like to see the experimental results. If not, I would like to see discussion on why the accurate momentum is useful for adversarial training but not useful for regular training.

---

### Official Review · Reviewer_WpNN · 2022-10-25

**Confidence:** 5
**Correctness:** 2
**Technical Novelty And Significance:** 1
**Empirical Novelty And Significance:** Not applicable
**Recommendation:** 3

**Clarity, Quality, Novelty And Reproducibility:**

Overall, the quality and originality of this paper are somehow not good. Please refer to my concerns in the *Weaknesses*. But it is easy to follow.

**Strength And Weaknesses:**

Strength
This paper did completely present the idea and is easy to follow.

Weaknesses
One of my biggest concerns is the unfair comparison in the experiment. The improved attacking performance of the generated adversarial examples on the target models is more likely caused by a stronger attack induced by “more iteration steps” (Steps 2 to 9 in algorithm 1). To me, it is not the reflection of the transferability of the generated adversarial examples. I suggest the experiment should be conducted under saturation attack and equally computational costs. Specifically, the computational costs of steps 2~9 in Algorithm 1 should be considered in other baselines, and a greater number of iterations should be used to validate the root cause of the performance improvement, attack strength, or attack transferability.
Another aspect of unfair comparison is the selected baselines. The selected baselines in the experiment seem to be a weakened version of the proposed method. Some parallel work should be used for a fair comparison. Besides, the work is claimed to be a work in the line of black-box attack, then some work in this line should be considered too [4].

My second concern is about the careless statements, some are listed as (1), (2).
(1)	In Paragraph 2 of the Introduction, “Iterative gradient-based and optimization-based attacks have high white-box but low black-box attack success rates, which means that such two attacks are impracticable in the real world”. Impracticable is not a reasonable conclusion for white-box attack. Please refer [3].
(2)	2.2 in the Related Work – the well-known fact about adversarial training is that insufficient inner maximization steps don’t induce enough robustness. However, adversarial training and its variants [1,2] with sufficient inner steps are still the golden baseline to defend against adversarial attacks. The statements about adversarial training are inaccurate and incomplete.

Another concern is the mathematical foundation of the work. The adversarial perturbation implies the local incremental of loss landscape over a specific point w.r.t a specific function. Setting initial values of perturbation to accelerate optimization is hard to corelated with the transferability of adversarial attacks.
Specifically, the statements “leading to better acceleration in the first few iterations”, and “accelerate optimization effectively during the early iterations to improv transferability”, do not make too much sense to me.

My last concern is about the novelty of the proposed method. The $g^{exp}$ in line 10, which is derived from line 2 to line 9 is the core idea of this paper, i.e., setting an initial value for the adversarial perturbation, which is not technically novel to me.

[1] Madry A, Makelov A, Schmidt L, et al. Towards deep learning models resistant to adversarial attacks[J]. arXiv preprint arXiv:1706.06083, 2017.
[2] Zhang H, Yu Y, Jiao J, et al. Theoretically principled trade-off between robustness and accuracy[C]//International conference on machine learning. PMLR, 2019: 7472-7482.
[3] Carlini N, Athalye A, Papernot N, et al. On evaluating adversarial robustness[J]. arXiv preprint arXiv:1902.06705, 2019.
[4] Shukla S N, Sahu A K, Willmott D, et al. Simple and efficient hard label black-box adversarial attacks in low query budget regimes[C]//Proceedings of the 27th ACM SIGKDD Conference on Knowledge Discovery & Data Mining. 2021: 1461-1469.

**Summary Of The Paper:**

This paper studied the problem of transferability of adversarial examples. Specifically, it considered the setting of an input sample and two independent networks that well-behave on the sample, if the adversarial example derived from a specific neural network and the sample can fool the other network.
Building upon previous work that practically prevents the adversarial example overfitting on a specific network by introducing input transformations and momentum strategy into the optimization procedures of generating adversarial examples, this paper proposes to accelerate the optimization procedures by setting an initial value for the momentum item for the iterative updating. The initial value can be considered a good estimation of the adversarial perturbation, so it is generated by a procedure similar to common adversarial attack methods but involving more randomness to improve its adaption.
Considering the way of getting the initial value of momentum, the method proposed in this work actually involves more updating steps compared with other baselines.
In the experiment, this work compared the attacking performance of the proposed method with methods applying momentum, and the result shows that the adversarial examples generated by the proposed methods have better attacking performance on new models with varying network architectures.

**Summary Of The Review:**

After reviewing the paper, this work cannot persuade me that the proposed method can potentially improve the transferability of adversarial attacks under a fair comparison. Secondly, the proposed method is not technically novel to me. Besides, the introduction,  background, and terminology are less well-written.

---

> ### Author Response · Authors · 2022-11-10
> **Explanation and supplement of comparisons in the experiment**
>
> Thank you for the review!
>
> ---
>
> **Review1:** One of my biggest concerns is the unfair comparison in the experiment.
> The improved attacking performance of the generated adversarial examples on the target models is more
> likely caused by a stronger attack induced by “more iteration steps” (Steps 2 to 9 in algorithm 1).
>
> **Response1:** No. The iteration steps $T$ for adversarial examples is independent of steps 2 to 9 in Algorithm 1.
>
> - Firstly, the original image $x$ is **pre-perturbed** in steps 2 to 9.
> - Secondly, the algorithm conducts a critical operation in step 10: $x^{adv}_1 \gets x$. This operation initializes the adversarial example $x^{adv}_1$ to the original image $x$.
> - Thirdly, the algorithm only updates the adversarial example $x^{adv}_t$ $T$ times in steps 11 to 13
> (the iteration steps of all baselines are also set to T).
> For each iteration, the direction is the sign of the momentum, and the step size is $alpha$.
>
> In general, the effect of steps 2 to 9 is to give the momentum an initial value ($g_0 \gets g^{exp}$, in step 10),
> which gives the momentum more iterations, but does not affect the iteration steps for the adversarial sample.
> Therefore, our comparison is fair.
>
> ---
>
> **Review2:** Another aspect of unfair comparison is the selected baselines.
> The selected baselines in the experiment seem to be a weakened version of the proposed method.
> Some parallel work ...
>
> **Response2:** In this paper, we have made the following comparisons to demonstrate that our EPN is better than the
> conventional momentum (Polyak and Nesterov momentum) in improvement of transferability:
>
> - Polyak or Nesterov momentum based attack [1, 2] *vs.* EPN-based attack
> - Polyak or Nesterov momentum based attack + input transformations [3, 4, 2] *vs.* EPN-based attack + input transformations
> - Polyak or Nesterov momentum based attack + variance tuning [5] *vs.* EPN-based attack + variance tuning
> - Polyak or Nesterov momentum based attack + feature importance-aware [6] *vs.* EPN-based attack + feature importance-aware
>
> We considered more comparisons of ensemble attacks (e.g., Polyak or Nesterov momentum based attack + variance tuning + input transformations *vs.* EPN-based attack + variance tuning + input transformations), but eventually discarded these experiments.
> The combination of methods will naturally improve the transferability further,
> but this does not better help us to demonstrate that our EPN is better than the conventional momentum in improvement of transferability.
> Therefore, according to Occam's razor principle, we choose the most basic baselines in the comparisons.
> However, we are willing to show more experiments in the paper under revision.
> Some experimental results are as follows:
>
> | Models | Attacks    | Inc-v3     | Inc-v4   | IncRes-v2 | Res-50   | Res-152  | Vgg-16   | Vgg-19   |
> |--------|---------|------------|----------|-----------|----------|----------|----------|----------|
> | Inc-v3 | VT-DTS-MI-FGSM              | 99.3*      | 73.7     | 68.7      | 74.8     | 70.5     | 77.0     | 77.6     |
> |        | VT-DTS-EPNI-FGSM (**Ours**) | **100.0*** | **88.1** | **83.1**  | **87.0** | **84.8** | **86.5** | **89.4** |
>
> | Models | Attacks   | Adv-Inc-v3 | Ens-IncRes-v2 | R&P      | Bit-Red  | JPEG     | FD       |
> |--------|-----------|------------|---------------|----------|----------|----------|----------|
> | Inc-v3 | VT-DTS-MI-FGSM              | 69.7       | 57.7          | 55.9     | 59.3     | 66.6     | 60.8     |
> |        | VT-DTS-EPNI-FGSM (**Ours**) | **79.3**   | **71.8**      | **74.2** | **74.5** | **82.4** | **77.9** |
>
> where DTS denotes the ensemble of DIM [3], TIM [4], and SIM [2].
> The experimental results show that our EPN also shows better performance in improvement of transferability.
>
> Besides, we focus on transferability-based black-box attacks, so we do not consider gradient estimation based
> and optimization-based black-box attacks as baselines.
>
> ---
>
> [1] Dong et al. Boosting adversarial attacks with momentum. In Proceedings of the IEEE conference on computer vision and pattern recognition, 2018.
>
> [2] Lin et al. Nesterov accelerated gradient and scale invariance for adversarial attacks. In International Conference on Learning Representations, 2020.
>
> [3] Xie et al. Improving transferability of adversarial examples with input diversity. In Proceedings of the IEEE/CVF Conference on Computer Vision and Pattern Recognition, 2019.
>
> [4] Dong et al. Evading defenses to transferable adversarial examples by translation-invariant attacks. In Proceedings of the IEEE/CVF Conference on Computer Vision and Pattern Recognition, 2019.
>
> [5] Wang et al. Enhancing the transferability of adversarial attacks through variance tuning. In Proceedings of the IEEE/CVF Conference on Computer Vision and Pattern Recognition, 2021.
>
> [6] Wang et al. Feature importance-aware transferable adversarial attacks. In Proceedings of the IEEE/CVF International
> Conference on Computer Vision, 2021.

---

> > ### Comment · Reviewer_WpNN · 2022-11-25
> > **Acknowledgement of Rebuttal**
> >
> > I would like to thank the authors for the detailed feedback and additional results.
> > 1) After reading the response, I still feel that the proposed would have more iterations since steps 2-9 are not independent in Algorithm 1 but rather sample related. Especially, `g^{exp}` depends on a specific input, and therefore the whole procedure counts for the computational cost.
> > 2) For the concern of selection of baseline methods. I expected methods beyond the scope of momentum based. The newly posted results partially solved my concerns about the selection of baselines since the proposed methods will reinforce the other methods instead of worsening the performance.
> > I would retain my original assessment based on the points above.

---

> > > ### Author Response · Authors · 2022-12-09
> > > **Discussion of more iterations and baselines**
> > >
> > > Thanks to the reviewer for the rebuttal.
> > >
> > > ---
> > >
> > > 1. If more iterations related to the example is impermissible or unfair, is the iterations of aggregate gradient related to the example also impermissible or unfair in FIA? Is it also impermissible or unfair that additional iterations related to the example are involved to obtain the gradient variance in VMI-FGSM? Is it the same for obtaining the gradients over the scale copies of the input image in SIM?
> > >
> > > 2. We aim to demonstrate that the proposed EPN is more effective than conventional momentum in the improvement of the transferability of the adversarial eamples, so we compare the proposed EPN-based attacks and conventional momentum-based attacks. However, what conclusions can be drawn from the comparison of the proposed EPN-based attacks and non-momentum-based attacks?

---

> ### Author Response · Authors · 2022-11-10
> **Careless statements and important explanation**
>
> **Review3:** My second concern is about the careless statements.
>
> **Response3:** Sorry for making mistakes in the statement.
> We are reviewing and revising the paper and will submit a new version of the paper shortly.
>
> ---
>
> **Review4:** Setting initial values of perturbation to accelerate optimization is hard to corelated with the transferability of adversarial attacks.
>
> **Response4:** No. We set the initial value of the momentum instead of the perturbation. The initial value of the perturbation is 0, and step 10 of Algorithm 1 can verify this statement.

---

### Decision · Program_Chairs · 2023-01-20

**Decision:**

Reject

**Justification For Why Not Higher Score:**

Poor motivation, inadequate experiments.

**Justification For Why Not Lower Score:**

N/A

**Metareview: Summary, Strengths And Weaknesses:**

This work proposed a new initialization of the momentum based adversarial attack, to improve the transferability to target models. The reported results look good, but there are several important concerns from the reviewers, mainly including:
1. Lack of motivation: the authors didn't provide convincing explanation about why the proposed initialization could improve the adversarial transferability.
2. Incremental technical novelty.
3. Careless statement.
4. Inadequate experiments.

Although the authors made some efforts in the rebuttal, most above concerns were not well addressed.